



# Technical Note: Analytical Inversion of the Parametric Budyko Equations

Nathan G. F. Reaver[1,2], David A. Kaplan[2], Harald Klammler[2,3], and James W. Jawitz[4]

[1]Water Institute, University of Florida, Gainesville, Florida, USA.
[2]Engineering School of Sustainable Infrastructure and Environment (ESSIE), University of Florida, Gainesville, Florida, USA.
[3]Department of Geosciences, Federal University of Bahia, Salvador, Bahia, Brazil.
[4]Soil and Water Science Department, University of Florida, Gainesville, Florida, USA.

*Correspondence to*: Nathan G. F. Reaver (nreaver@ufl.edu)

**Abstract.** The non-parametric Budyko framework provides empirical relationships between a catchment's long-term mean
evapotranspiration ($\bar{E}$) and the aridity index, defined as the ratio of mean rainfall depth ($\bar{P}$) to mean potential evapotranspiration
($\overline{E_0}$). The parametric Budyko equations attempt to generalize this framework by introducing a catchment-specific parameter
($n$ or $w$), intended to represent differences in catchment climate and landscape features. Many studies have developed complex
regression relationships for the catchment-specific parameter in terms of biophysical features, all of which use known values
of $\bar{P}$, $\overline{E_0}$, and $\bar{E}$ to numerically invert the parametric Budyko equations to obtain values of $n$ or $w$. In this study, we analytically
invert both forms of the parametric Budyko equations, producing expressions for $n$ and $w$ only in terms of $\bar{P}$, $\overline{E_0}$, and $\bar{E}$. These
expressions allow for $n$ and $w$ to be explicitly expressed in terms of biophysical features through the dependence of $\bar{P}$, $\overline{E_0}$, and
$\bar{E}$ on those same features.

## 1 Introduction

The non-parametric Budyko framework was developed to explain and describe the distinctive clustering pattern
observed for the long-term average evaporative behavior across multiple catchments. This pattern emerges when the
evaporative indices, $\frac{\bar{E}}{\bar{P}}$ (where $\bar{P}$ is the mean rainfall depth and $\bar{E}$ is the mean actual evapotranspiration depth), of multiple
catchments are plotted against their corresponding aridity indices, $\frac{\overline{E_0}}{\bar{P}}$ (where $\overline{E_0}$ is the mean potential evapotranspiration
depth). Several empirical relationships of the form,

$$\frac{\bar{E}}{\bar{P}} = f_0\left(\frac{\overline{E_0}}{\bar{P}}\right),\tag{1}$$

have been proposed to describe this pattern, including (Schreiber, 1904),

$$\frac{\bar{E}}{\bar{P}} = 1 - e^{-\frac{\overline{E_0}}{\bar{P}}},\tag{2}$$

and (Ol'Dekop, 1911)



$$\frac{\bar{E}}{\bar{P}} = \frac{\overline{E_0}}{\bar{P}} \tanh\left(\frac{\bar{P}}{\overline{E_0}}\right). \tag{3}$$

Equations (2) and (3) were selected since they closely match the central tendency of $\frac{\bar{E}}{\bar{P}}$ across $\frac{\overline{E_0}}{\bar{P}}$, obey the laws of conservation

of energy and mass for all values of $\frac{\overline{E_0}}{\bar{P}}$, and approach energy limitation (i.e., $\frac{\bar{E}}{\bar{P}} \to \frac{\overline{E_0}}{\bar{P}}$) and water limitation (i.e., $\frac{\bar{E}}{\bar{P}} \to 1$) in the

humid (i.e., $\frac{\overline{E_0}}{\bar{P}} \to 0$) and arid (i.e., $\frac{\overline{E_0}}{\bar{P}} \to \infty$) limits, respectively. The geometric mean of Eq. (2) and (3) has been shown to

predict $\frac{\bar{E}}{\bar{P}}$ with ~10% uncertainty (Budyko and Zubenok, 1961; Gentine et al., 2012) for ungauged basins if $\bar{P}$ and $\overline{E_0}$ are known.

In an attempt to generalize the non-parametric Budyko framework and explain deviations in $\frac{\bar{E}}{\bar{P}}$ from the central

tendency of the empirically observed catchment clustering pattern, two parametric Budyko equations have been proposed

(Turc, 1953; Choudhury, 1999; Mezentsev, 1955; Yang et al., 2008),

$$\frac{\bar{E}}{\bar{P}} = \frac{\frac{\overline{E_0}}{\bar{P}}}{\left[1 + \left(\frac{\overline{E_0}}{\bar{P}}\right)^n\right]^{\frac{1}{n}}}, \tag{4}$$

and (Tixeront, 1964; Berkaloff and Tixeront, 1958; Fu, 1981; Zhang et al., 2004),

$$\frac{\bar{E}}{\bar{P}} = 1 + \frac{\overline{E_0}}{\bar{P}} - \left(1 + \left(\frac{\overline{E_0}}{\bar{P}}\right)^w\right)^{\frac{1}{w}}, \tag{5}$$

where $n$ and $w$ are dubbed "catchment-specific parameters". Equations (4) and (5) can alternatively and equivalently be

expressed in terms of the R-Index, $\frac{\bar{E}}{\overline{E_0}}$ (Yao, 1974), and humidity index, $\frac{\bar{P}}{\overline{E_0}}$ (Hulme et al., 1992), giving,

$$\frac{\bar{E}}{\overline{E_0}} = \frac{\frac{\bar{P}}{\overline{E_0}}}{\left[1 + \left(\frac{\bar{P}}{\overline{E_0}}\right)^n\right]^{\frac{1}{n}}}, \tag{6}$$

and

$$\frac{\bar{E}}{\overline{E_0}} = 1 + \frac{\bar{P}}{\overline{E_0}} - \left(1 + \left(\frac{\bar{P}}{\overline{E_0}}\right)^w\right)^{\frac{1}{w}}. \tag{7}$$

The catchment-specific parameter ($n$ or $w$) has been described as an empirical, effective parameter representing the

influence of all catchment biophysical features, other than $\bar{P}$ and $\overline{E_0}$, on $\bar{E}$ (Wang et al., 2016a). Additionally, the functional

forms of the parametric Budyko equations have typically been interpreted as representing the evaporative behavior of

individual catchments under different aridity indices (e.g., (Roderick and Farquhar, 2011; Wang and Hejazi, 2011; Yang and

Yang, 2011; Wang et al., 2016b; Zhou et al., 2016; Shen et al., 2017; Zhang et al., 2016; Milly et al., 2018)). Utilizing these

interpretations, many studies have developed complex regression relationships for the catchment-specific parameter in terms

of various climate and landscape features (Yang et al., 2007; Donohue et al., 2012; Yang et al., 2009; Shao et al., 2012; Li et





al., 2013; Xu et al., 2013; Cong et al., 2015; Yang et al., 2016; Zhang et al., 2018; Abatzoglou and Ficklin, 2017; Xing et al., 2018; Zhao et al., 2020; Ning et al., 2020b; Ning et al., 2020a; Li et al., 2020b; Li et al., 2020a; Zhang et al., 2019; Ning et al., 2019; Bai et al., 2019; Ning et al., 2017). In all such studies, known values of $\bar{P}$, $\overline{E_0}$, and $\bar{E}$, estimated empirically or via modelling, are used to numerically invert the parametric Budyko equations to obtain values of the catchment-specific

parameter, which are then regressed against various biophysical features. The expressions obtained from such endeavours vary considerably between studies, both in their functional forms and which biophysical features are included, making it difficult to develop a consistent mechanistic understanding of the catchment-specific parameter (Reaver et al., 2020). This difficulty could be overcome by having the explicit relationship between $n$ or $w$ and $\bar{P}$, $\overline{E_0}$, and $\bar{E}$, which would allow for $n$ or $w$ to be expressed in terms of biophysical features through the dependence of $\bar{P}$, $\overline{E_0}$, and $\bar{E}$ on those same features.

In this study, we analytically invert both forms of the parametric Budyko equations. The resulting expressions give $n$ and $w$ only in terms of $\bar{P}$, $\overline{E_0}$, and $\bar{E}$, illustrating that if $n$ and $w$ depend on any biophysical features, it is due directly to the dependence of $\bar{P}$, $\overline{E_0}$, or $\bar{E}$ on those same features. Notably, there has not been an analytical derivation illustrating how $n$ and $w$ relate to biophysical features, though the importance of doing so has been noted many times (Zhang et al., 2004; Yang et al., 2008; Donohue et al., 2012; Xu et al., 2013; Greve et al., 2015; Wang et al., 2016a; Zhang et al., 2018). The expressions

we develop here for $n$ and $w$ satisfy this need, providing a general expression for the dependence of $n$ and $w$ on any possible biophysical features through the dependence of $\bar{P}$, $\overline{E_0}$, and $\bar{E}$ on those same features.

## 2 Analytical Expressions for $n$ and $w$

To develop explicit analytical expressions for $n$ and $w$, we start with Eq. (4) and (5). These two equations can be algebraically manipulated into,

$$(e^n)^{\ln\left(\frac{\overline{E_0}}{\bar{E}}\right)} - (e^n)^{\ln\left(\frac{\overline{E_0}}{\bar{P}}\right)} = 1 \,, \tag{8}$$

and

$$(e^w)^{\ln\left(\frac{\overline{E_0}+\bar{P}-\bar{E}}{\bar{P}}\right)} - (e^w)^{\ln\left(\frac{\overline{E_0}}{\bar{P}}\right)} = 1 \,, \tag{9}$$

respectively. Similarly, Eq. (6) and (7) can be manipulated into

$$(e^n)^{\ln\left(\frac{\bar{P}}{\bar{E}}\right)} - (e^n)^{\ln\left(\frac{\bar{P}}{\overline{E_0}}\right)} = 1 \,, \tag{10}$$

and

$$(e^w)^{\ln\left(\frac{\overline{E_0}+\bar{P}-\bar{E}}{\overline{E_0}}\right)} - (e^w)^{\ln\left(\frac{\bar{P}}{\overline{E_0}}\right)} = 1 \,, \tag{11}$$

respectively. Eq. (8), (9), (10), and (11) all have the following general form,


$$y^C + zy^D = 1 \,, \tag{12}$$

where $C$ and $D$ are constants related to the evaporative and aridity indices, respectively, $z$ is an arbitrary complex variable, and

$y$ is a function of $z$. With the constraint, $C > D > 0$, Eq. (12) has a solution of the form (Hochstadt, 2012), p. 81-84,

$$y(z) = \frac{1}{C} \sum_{r=0}^{\infty} \frac{(-1)^r z^r \Gamma\left(\frac{1+Dr}{C}\right)}{r! \Gamma\left(\frac{1+Dr}{C}+1-r\right)} \,, \tag{13}$$

where $r$ is an integer index variable, and $\Gamma(\ )$ is the gamma function. Table 1 illustrates that Eq. (8) and (9) fulfill the necessary

constraints (i.e. $C > D > 0$) for arid climates (i.e. $\overline{E_0} > \bar{P}$) and Eq. (10) and (11) do so for humid climates (i.e. $\overline{E_0} < \bar{P}$), which

leads to the explicit analytical expression for $n$ and $w$,

$$n \text{ or } w \ = \ln\left[\frac{1}{\ln(G)} \sum_{r=0}^{\infty} \frac{\Gamma\left(\frac{1+\ln(H)r}{\ln(G)}\right)}{\Gamma(r+1)\Gamma\left(\frac{1+\ln(H)r}{\ln(G)}+1-r\right)}\right] \,, \tag{14}$$

where $G_n = \left\{\frac{\overline{E_0}}{\bar{E}}, \frac{\bar{P}}{\bar{E}}\right\}$, $G_w = \left\{\frac{\overline{E_0}+\bar{P}-\bar{E}}{\bar{P}}, \frac{\overline{E_0}+\bar{P}-\bar{E}}{\overline{E_0}}\right\}$, and $H_{n,w} = \left\{\frac{\overline{E_0}}{\bar{P}}, \frac{\bar{P}}{\overline{E_0}}\right\}$; the first and second terms inside the braces apply to arid

and humid climates, respectively (see also explicit forms in Eq. (A47)-(A50) of appendix). For critical point climates, where

$\overline{E_0} = \bar{P}$, Eq. (8), (9), (10), and (11) can be solved algebraically, giving,

$$n \text{ or } w \ = \frac{\ln(2)}{\ln(\Omega)} \,, \tag{15}$$

where $\Omega_n = \frac{\overline{E_0}}{\bar{E}} = \frac{\bar{P}}{\bar{E}}$ and $\Omega_w = 2 - \frac{\bar{E}}{\overline{E_0}} = 2 - \frac{\bar{E}}{\bar{P}}$ (see also Eq. (A51) and (A52) of appendix). The detailed derivations of Eq. (14)

and (15), are presented in Appendix A.





**Table 1: Illustration of the condition $C > D > 0$ for arid climates (Eq. (8) and (9)) and for humid climates (Eq. (10) and (11)). From Left to Right: The first column gives the equation and associated climate. The second column gives mathematical constraints that must be true given an arid or humid climate. The third column gives specific mathematical constraints derived from the climatic constraints. The last column gives the condition, $C > D > 0$, for each associated equation and climate, given the climatic and derived constraints.**

| Version of Eq. (14) and Climate | Climatic Constraint | Derived Constraints | Resulting Condition, $C > D > 0$ |
|---|---|---|---|
| $n$ version, arid | $\overline{E_0} > \bar{P} > \bar{E}$ | $\dfrac{\overline{E_0}}{\bar{E}} > \dfrac{\overline{E_0}}{\bar{P}} > 1$ | $\ln\left(\dfrac{\overline{E_0}}{\bar{E}}\right) > \ln\left(\dfrac{\overline{E_0}}{\bar{P}}\right) > 0$ |
| $n$ version, humid | $\bar{P} > \overline{E_0} > \bar{E}$ | $\dfrac{\bar{P}}{\bar{E}} > \dfrac{\bar{P}}{\overline{E_0}} > 1$ | $\ln\left(\dfrac{\bar{P}}{\bar{E}}\right) > \ln\left(\dfrac{\bar{P}}{\overline{E_0}}\right) > 0$ |
| $w$ version, arid | $\overline{E_0} > \bar{P} > \bar{E}$ | $\dfrac{\overline{E_0} + \bar{P} - \bar{E}}{\bar{P}} > \dfrac{\overline{E_0}}{\bar{P}} > 1$ | $\ln\left(\dfrac{\overline{E_0} + \bar{P} - \bar{E}}{\bar{P}}\right) > \ln\left(\dfrac{\overline{E_0}}{\bar{P}}\right) > 0$ |
| $w$ version, humid | $\bar{P} > \overline{E_0} > \bar{E}$ | $\dfrac{\bar{P} + \overline{E_0} - \bar{E}}{\overline{E_0}} > \dfrac{\bar{P}}{\overline{E_0}} > 1$ | $\ln\left(\dfrac{\bar{P} + \overline{E_0} - \bar{E}}{\overline{E_0}}\right) > \ln\left(\dfrac{\bar{P}}{\overline{E_0}}\right) > 0$ |

## 3 Properties of Analytical Expressions for $n$ and $w$

Here we investigate the mathematical properties of Eq. (14) and (15) to determine that they are valid analytical expressions for $n$ and $w$. First, we examine the behavior of Eq. (15) as $\bar{E} \to 0$ and $\bar{E} \to \overline{E_0}$ or $\bar{P}$. Mathematically, the values of $n$ are constrained between 0 and $\infty$ (Yang et al., 2008), and the values of $w$ are constrained between 1 and $\infty$ (Zhang et al., 2004). Therefore, the upper and lower limits of the $n$ and $w$ versions of Eq. (15) should be equal to these respective constraints. The lower limit for the $n$ version of Eq. (15),

$$\lim_{\bar{E} \to 0} n = \frac{\ln(2)}{\infty} = 0 \,, \tag{16}$$

and the upper limit,

$$\lim_{\bar{E} \to \overline{E_0}} n = \lim_{\bar{E} \to \bar{P}} n = \frac{\ln(2)}{0} = \infty \,, \tag{17}$$




are equal to the lower and upper constraint for $n$, respectively. Similarly, the lower limit for the $w$ version of Eq. (15),

$$\lim_{\bar{E}\to 0} w = \frac{\ln(2)}{\ln(2-0)} = 1 \; ,$$
(18)

and the upper limit,

$$\lim_{\bar{E}\to \overline{E_0}} n = \lim_{\bar{E}\to \bar{P}} n = \frac{\ln(2)}{\ln(2-1)} = \frac{\ln(2)}{0} = \infty \; ,$$
(19)

are equal to the lower and upper constraint for $w$, respectively.

Next, we investigate the properties of Eq. (14). This equation contains a convergent infinite series whose value asymptotically approaches $e^n$ or $e^w$ for the $n$ and $w$ versions, respectively. The asymptotic behavior of the series' terms (e.g., monotonically decreasing or alternating sign and absolute value decreasing) depends on the specific values of $\bar{E}, \bar{P}$, and $\overline{E_0}$ (Fig. 1-4). To verify that both the $n$ and $w$ versions of Eq. (14) produce the correct values of $n$ and $w$ for a given set of $\bar{E}, \bar{P}$, and $\overline{E_0}$, we numerically invert Eq. (4) and (5) and compare the fitted $n$ and $w$ values, $\check{n}$ and $\check{w}$, to successively better approximations of Eq. (14). The numerical inversion of Eq. (4) and (5) consists of numerically solving,

$$\left[ \frac{\frac{\overline{E_0}}{\bar{P}}}{\left[1+\left(\frac{\overline{E_0}}{\bar{P}}\right)^{\check{n}}\right]^{\frac{1}{\check{n}}}} - \frac{\bar{E}}{\bar{P}} \right]^2 = 0 \; .$$
(20)

and

$$\left[ 1 + \frac{\overline{E_0}}{\bar{P}} - \left(1+\left(\frac{\overline{E_0}}{\bar{P}}\right)^{\check{w}}\right)^{\frac{1}{\check{w}}} - \frac{\bar{E}}{\bar{P}} \right]^2 = 0 \; .$$
(21)

for $\check{n}$ and $\check{w}$, respectively. We compute approximations of Eq. (14) by truncating the infinite series to a finite number of terms, $N_r$. We successively improve these finite approximations by increasing $N_r$. As the number of terms in the finite series increase, the approximations asymptotically converge to the $\check{n}$ and $\check{w}$ values obtained from the numerical inversion of Eq. (4) and (5) for both arid and humid climates (Fig. 1-4). This convergence is rapid (requiring fewer than ten terms) for typical values of $n$ and $w$ (i.e., $< 4$) and provides strong numeric evidence that Eq. (14) yields valid analytical expressions for $n$ and $w$.

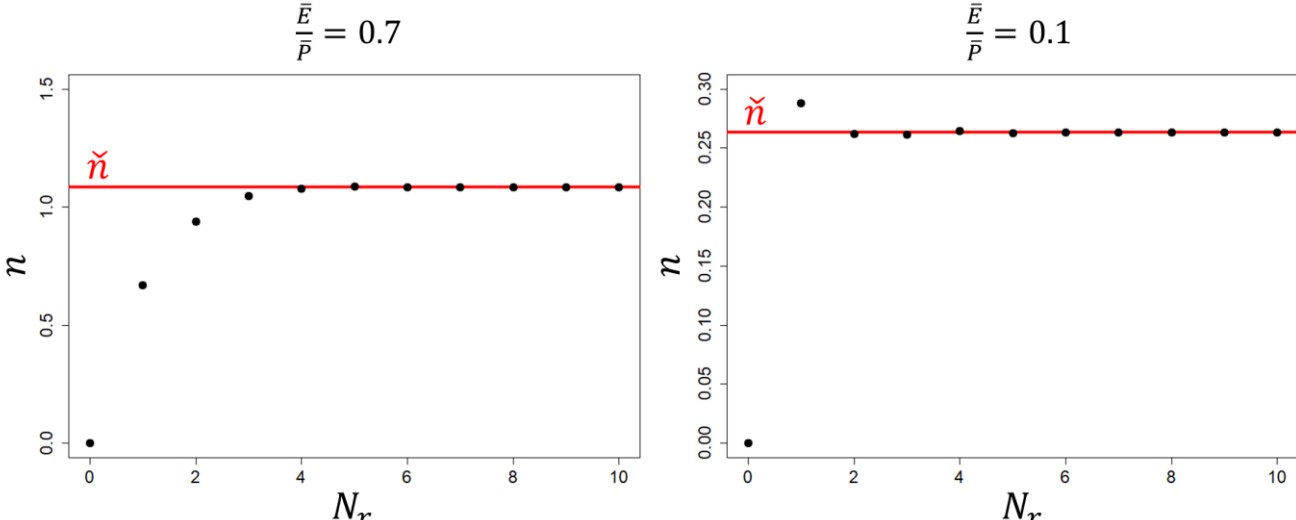

**Figure 1: Comparison of $\check{n}$ (red lines) from Eq. (20) to finite series approximations of $n$ (black dots) from Eq. (14) for increasing values of $N_r$ and an arid catchment $\left(\frac{\overline{E_0}}{\overline{P}} = 2\right)$, with $\frac{\overline{E}}{\overline{P}} = 0.7$ (left panel) and $\frac{\overline{E}}{\overline{P}} = 0.1$ (right panel). In both cases, the finite series approximations of $n$ asymptotically converge to $\check{n}$ with increasing numbers of terms.**

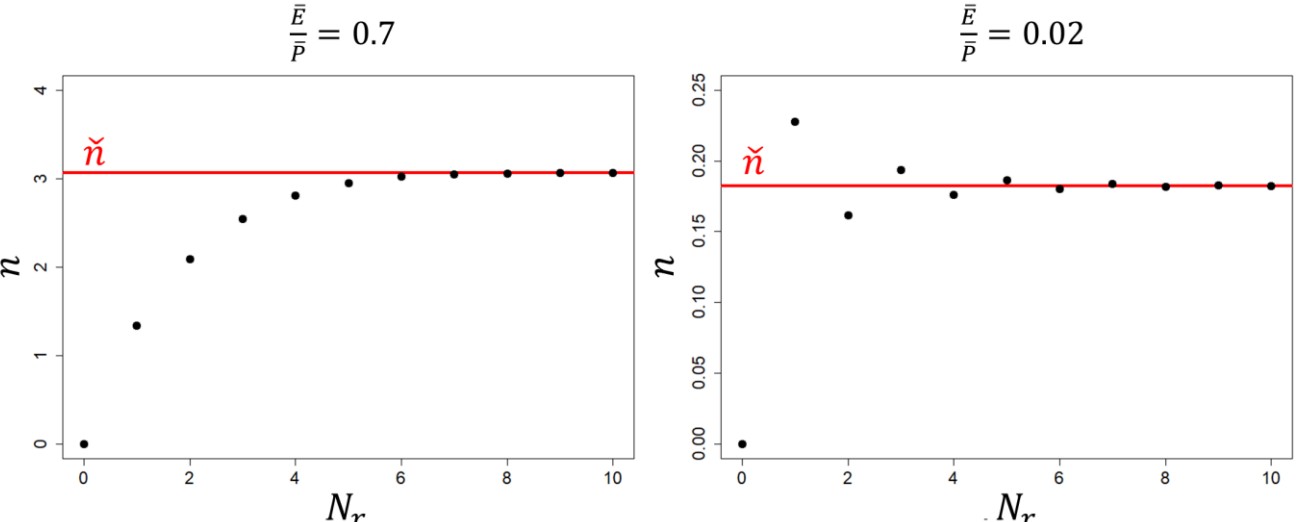

**Figure 2: Comparison of $\check{n}$ (red lines) from Eq. (20) to finite series approximations of $n$ (black dots) from Eq. (14) for increasing values of $N_r$ and a humid catchment $\left(\frac{\overline{E_0}}{\overline{P}} = 0.8\right)$, with $\frac{\overline{E}}{\overline{P}} = 0.7$ (left panel) and $\frac{\overline{E}}{\overline{P}} = 0.02$ (right panel). In both cases, the finite series approximations of $n$ asymptotically converge to $\check{n}$ with increasing numbers of terms.**





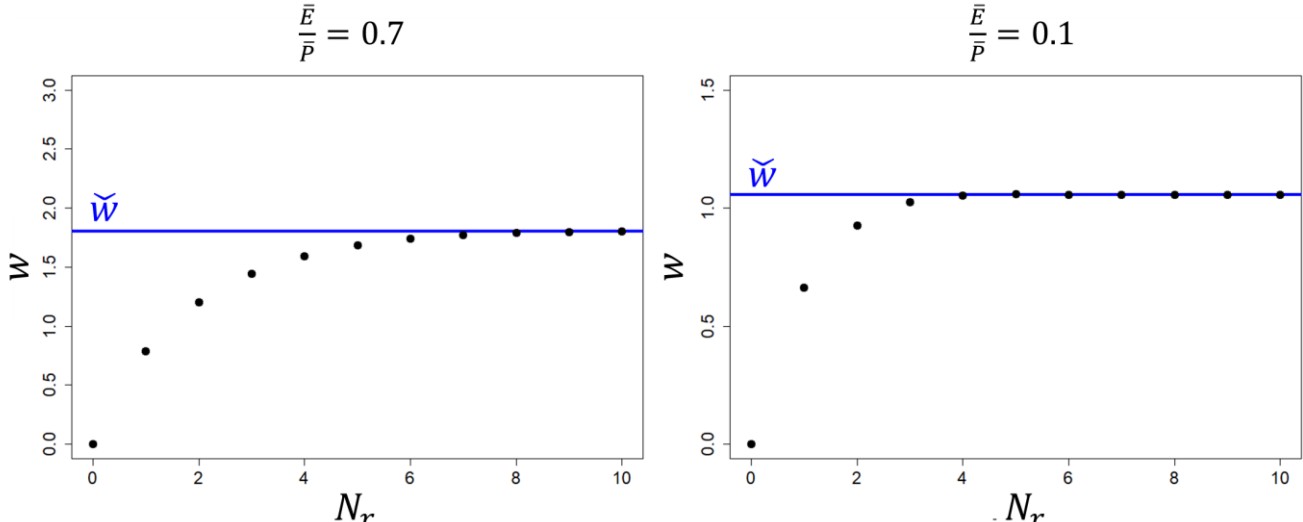

**Figure 3: Comparison of $\breve{w}$ (blue lines) from Eq. (21) to finite series approximations of $w$ (black dots) from Eq. (14) for increasing values of $N_r$ and an arid catchment $\left(\frac{\overline{E_0}}{\overline{P}} = 2\right)$, with $\frac{\overline{E}}{\overline{P}} = 0.7$ (left panel) and $\frac{\overline{E}}{\overline{P}} = 0.1$ (right panel). In both cases, the finite series approximations of $w$ asymptotically converge to $\breve{w}$ with increasing numbers of terms.**

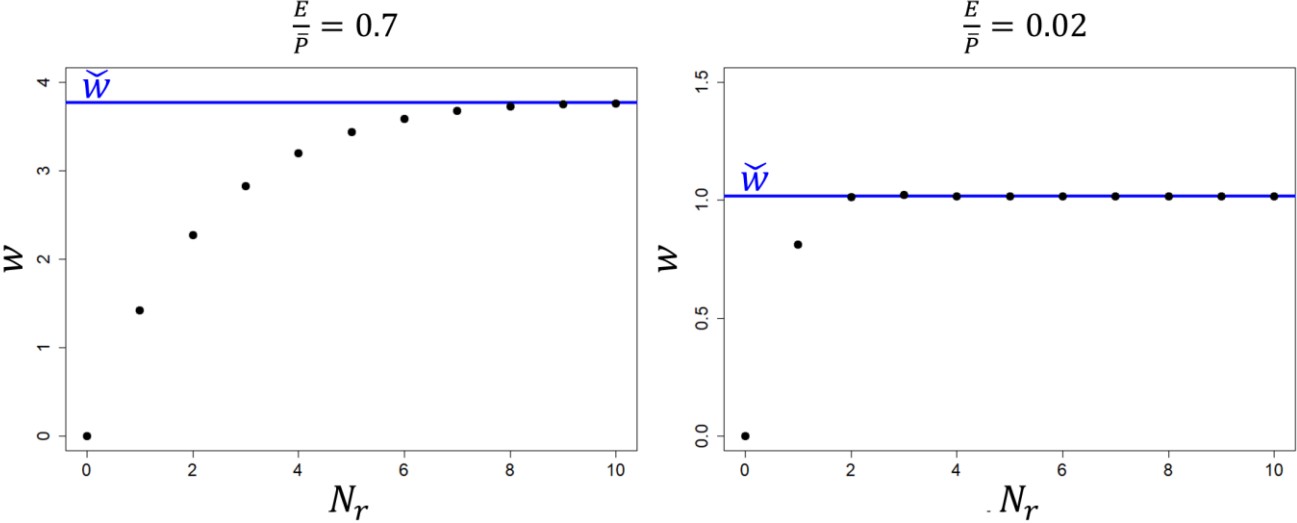

**Figure 4: Comparison of $\breve{w}$ (blue lines) from Eq. (21) to finite series approximations of $w$ (black dots) from Eq. (14) for increasing values of $N_r$ and a humid catchment $\left(\frac{\overline{E_0}}{\overline{P}} = 0.8\right)$, with $\frac{\overline{E}}{\overline{P}} = 0.7$ (left panel) and $\frac{\overline{E}}{\overline{P}} = 0.02$ (right panel). In both cases, the finite series approximations of $w$ asymptotically converge to $\breve{w}$ with increasing numbers of terms.**


## 4 Discussion and Conclusion

Inspection of Eq. (14) highlights the ambiguous nature of $n$ and $w$. The relationships between $n$, $w$, $\bar{E}$, $\overline{E_0}$, and $\bar{P}$ are fairly complicated, including logarithms of infinite series for the non-redundant cases. This highly nonlinear relationship challenges an intuitive understanding of how $n$ and $w$ should vary with changes to $\bar{E}$, $\overline{E_0}$, or $\bar{P}$ and is likely part of the reason

why attempts to relate the catchment-specific parameter to climate and landscape features have yielded such divergent results (e.g., (Yang et al., 2007; Donohue et al., 2012; Yang et al., 2009; Shao et al., 2012; Li et al., 2013; Xu et al., 2013; Cong et al., 2015; Yang et al., 2016; Zhang et al., 2018; Abatzoglou and Ficklin, 2017; Xing et al., 2018; Zhao et al., 2020; Ning et al., 2020b; Ning et al., 2020a; Li et al., 2020b; Li et al., 2020a; Zhang et al., 2019; Ning et al., 2019; Bai et al., 2019; Ning et al., 2017)).

Notably, the explicit analytical expression for $n$ and $w$ from Eq. (14) illustrates that the value of the catchment-specific parameter is only determined by $\bar{E}$, $\overline{E_0}$, and $\bar{P}$. Therefore, if $n$ or $w$ depend on biophysical features, it is directly due to the dependence of $\bar{E}$, $\overline{E_0}$, or $\bar{P}$ on those features. In short, this means that Eq. (14) is the general solution for how $n$ and $w$ depend on biophysical features. By substituting $\bar{E}$, $\overline{E_0}$, or $\bar{P}$ as functions of specific biophysical features into Eq. (14), one obtains the expression for $n$ and $w$ as a function of those features. Eq. (14) thus fulfills the literature-identified need of an

analytical expression for $n$ and $w$ in terms of biophysical features.

**Author contributions**

NGFR conceived the study, performed the analytical inversion and analyses, and drafted the manuscript. All authors contributed in the interpretation of results and manuscript preparation.

**Competing interests**

The authors declare no conflicts of interest with respect to the results of this manuscript.

**Acknowledgments**

NGFR would like to acknowledge support from the University of Florida Graduate Fellowship.

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

## Appendix A: Details of Analytical Inversion

Here we develop analytical inversions of the two parametric Budyko equations. The steps for these derivations are:

1) Produce a general form for both parametric forms of the Budyko equation.

2) Determine the Mellin transform for the general form.

3) Manipulate the Mellin transformed general form into functions with known inversions.

4) Take the inverse Mellin transform of the general form to find a general solution.

5) Substitute the specific functional forms of each parametric Budyko equation into the general solution to produce explicit expressions for $n$ and $w$.

The outline for portions of these derivations was developed from Hochstadt (2012), p. 81-84.

### A.1 Producing a general form for both parametric forms of the Budyko equation

Equation (4) of the main text can be rearranged in the following manner,

$$\left(\frac{E_0}{E}\right)^n - \left(\frac{E_0}{P}\right)^n = \left(e^{\ln\left(\frac{E_0}{E}\right)}\right)^n - \left(e^{\ln\left(\frac{E_0}{P}\right)}\right)^n = e^{n\ln\left(\frac{E_0}{E}\right)} - e^{n\ln\left(\frac{E_0}{P}\right)} = (e^n)^{\ln\left(\frac{E_0}{E}\right)} - (e^n)^{\ln\left(\frac{E_0}{P}\right)} = 1 , \qquad (A1)$$





to produce Eq. (8). Following the same procedure, Eq. (5), (6), and (7) can be rearranged to Eq. (9), (10), and (11). The general form of Eq. (8), (9), (10), and (11) is Eq. (12). We use the Eq. (12) and the condition, $C > D > 0$ (Table 1), to derive a general solution which will allow for specific solutions of $n$ and $w$.

### A.2 The Mellin transform for $y(z)$

The Mellin transform for $y(z)$ of Eq. (12) is,

$$Y(s) = \int_0^\infty z^{s-1} y(z) dz \,, \tag{A2}$$

where $s$ is a complex number. Whether the improper integral in Eq. (A2) converges or diverges depends on the behavior of $y(z)$ and the value of $s$. Letting $z = 0$ in Eq. (12) gives,

$$y(0) = 1 \,. \tag{A3}$$

Taking the first derivative of Eq. (12) gives,

$$\frac{dy}{dz} = \frac{-y^D}{Cy^{C-1} + zDy^{D-1}} \,, \tag{A4}$$

Since $C > D > 0$, Eq. (A4) is always negative, meaning that $y(z)$ is a monotonically decreasing function for $0 \leq z < \infty$. Additionally, $y(z) = 0$ is not a valid solution to Eq. (12), therefore, $1 \geq y(z) > 0$ for $0 \leq z < \infty$. As $z \to \infty$, $y(z)$ will become very small. This allows us to rearrange Eq. (12) to get an approximate functional form for $y(z)$ for large values of $z$,

$$y^D = \frac{1 - y^C}{z} \approx \frac{1}{z} \,, \tag{A5}$$

therefore,

$$y(z) \approx z^{-\frac{1}{D}}. \tag{A6}$$

Thus, from Eq. (A3), (A4), and (A6), the integrand of Eq. (A2) is $\approx z^{s-1}$ for small values of $z$ and transitions to $\approx z^{s-\frac{1}{D}-1}$ for as $z$ increases. Splitting Eq. (A2) into two component integrals and substituting in Eq. (A3) for small values of $z$ and Eq. (A6)

for large values of $z$ gives,

$$\int_0^\infty z^{s-1} y(z) dz = \int_0^1 z^{s-1} y(z) dz + \int_1^\infty z^{s-1} y(z) dz \approx \int_0^1 z^{s-1} dz + \int_1^\infty z^{s-\frac{1}{D}-1} dz \,. \tag{A7}$$

To find the values of $s$ for which Eq. (A2) converges, we determine the values of $s$ for which the two approximate integrals in Eq. (A7) converge. Integrating Eq. (A7), and expressing $s$ as the sum of its real and imaginary parts, $s = a + bi$ where $i = \sqrt{-1}$, yields,

$$\int_0^1 z^{s-1} dz + \int_1^\infty z^{s-\frac{1}{D}-1} dz = \left[\frac{z^s}{s}\right]_0^1 + \left[\frac{z^{s-\frac{1}{D}}}{s-\frac{1}{D}}\right]_1^\infty = \left[\frac{z^{a+bi}}{a+bi}\right]_0^1 + \left[\frac{z^{a+bi-\frac{1}{D}}}{a+bi-\frac{1}{D}}\right]_1^\infty = \left[\frac{z^{bi} z^a}{a+bi}\right]_0^1 + \left[\frac{z^{bi} z^{a-\frac{1}{D}}}{a+bi-\frac{1}{D}}\right]_1^\infty \,. \tag{A8}$$

Rearranging Eq. (A8) and using Euler's formula, $e^{ix} = \cos(x) + i\sin(x)$, yields,



$$\left[\frac{e^{ib\ln(z)}z^a}{a+bi}\right]_0^1 + \left[\frac{e^{ib\ln(z)}z^{a-\frac{1}{D}}}{a+bi-\frac{1}{D}}\right]_1^\infty = \left[\frac{[\cos(b\ln(z))+i\sin(b\ln(z))]z^a}{a+bi}\right]_0^1 + \left[\frac{[\cos(b\ln(z))+i\sin(b\ln(z))]z^{a-\frac{1}{D}}}{a+bi-\frac{1}{D}}\right]_1^\infty . \tag{A9}$$

The value of the expression, $\cos(b\ln(z)) + i\sin(b\ln(z))$, in Eq. (A9) is bounded to a circular region of radius 1 around the origin in the complex plane, making it finite for all value of $z$. Evaluating Eq. (A9) and setting $F(z) = \cos(b\ln(z)) +$

$i\sin(b\ln(z))$ to represent its finite value, gives,

$$\left[\frac{F(z)z^a}{a+bi}\right]_0^1 + \left[\frac{F(z)z^{a-\frac{1}{D}}}{a+bi-\frac{1}{D}}\right]_1^\infty = \frac{F(1)}{a+bi} - \frac{F(0)}{a+bi}\lim_{z\to 0}z^a + \frac{F(\infty)}{a+bi-\frac{1}{D}}\lim_{z\to\infty}z^{a-\frac{1}{D}} - \frac{F(1)}{a+bi-\frac{1}{D}} . \tag{A10}$$

If both limits of Eq. (A10), $\lim_{z\to 0} z^a$ and $\lim_{z\to\infty} z^{a-\frac{1}{D}}$, exist, then Eq. (A2) will be convergent. The first of these limits, $\lim_{z\to 0} z^a$, will give a finite value if,

$$a > 0 , \tag{A11}$$

The second limit, $\lim_{z\to\infty} z^{a-\frac{1}{D}}$, will give a finite value if,

$$a - \frac{1}{D} < 0 , \tag{A12}$$

which gives,

$$a < \frac{1}{D} . \tag{A13}$$

Thus, Eq. (A2) converges and the Mellin transform exists if,

$$0 < Re(s) < \frac{1}{D} . \tag{A14}$$

### A.3 Mellin transformed $y(z)$ in terms of known functions

Next, we evaluate Eq. (A2) explicitly. To do this, we switch the integration from z to y, using Eq. (12). This involves expressing z in terms of $y$,

$$z = y^{-D} - y^{C-D} , \tag{A15}$$

expressing $dz$ in terms of y and $dy$,

$$dz = -[Dy^{-D-1} + (C-D)y^{C-D-1}]dy , \tag{A16}$$

and expressing the limits of integration in terms of $y$,

$$\begin{matrix} \text{when } z = 0, & y = 1 \\ \text{when } z \to \infty, & y \to 0 \end{matrix} . \tag{A17}$$

We can now rewrite Eq. (A2) in terms of a $y$ integration,



$\quad Y(s) = -\int_1^0 y(y^{-D} - y^{C-D})^{s-1}[Dy^{-D-1} + (C-D)y^{C-D-1}]dy$ , $\qquad$ (A18)

which can be rearranged to,

$$Y(s) = D\int_0^1 y^{-Ds}(1-y^C)^{s-1}dy + (C-D)\int_0^1 y^{-Ds+C}(1-y^C)^{s-1}dy .\qquad(A19)$$

Making the substitution $y^C = u$, we have,

$$dy = \frac{1}{C}y^{-C+1}du = \frac{1}{C}u^{\frac{1}{C}-1}du ,\qquad(A20)$$

$\quad$ while the limits of integration remain the same,

$$\begin{array}{ll} \text{when } y = 0, & u = 0 \\ \text{when } y = 1, & u = 1 \end{array},\qquad(A21)$$

therefore, Eq. (A19) becomes,

$$Y(s) = \frac{D}{C}\int_0^1 u^{\frac{1-Ds}{C}-1}(1-u)^{s-1}du + \frac{C-D}{C}\int_0^1 u^{\frac{1-Ds}{C}+1-1}(1-u)^{s-1}du .\qquad(A22)$$

The integrals in Eq. (A22) are of the same form as the integral definition of the Beta function, allowing $Y(s)$ to be expressed
$\quad$ as a sum of Beta functions,

$$Y(s) = \left[\frac{D}{C}\right]B\left(\frac{1-Ds}{C},s\right) + \left[\frac{C-D}{C}\right]B\left(\frac{1-Ds}{C}+1,s\right),\qquad(A23)$$

where $B(\ ,\ )$ is the Beta function. The Beta function in turn can also be defined in terms of the Gamma function (i.e. $B(q,L) = \frac{\Gamma(q)\Gamma(L)}{\Gamma(q+L)}$), allowing Eq. (A23) to be rewritten as,

$$Y(s) = \left[\frac{D}{C}\right]\frac{\Gamma\left(\frac{1-Ds}{C}\right)\Gamma(s)}{\Gamma\left(\frac{1-Ds}{C}+s\right)} + \left[\frac{C-D}{C}\right]\frac{\Gamma\left(\frac{1-Ds}{C}+1\right)\Gamma(s)}{\Gamma\left(\frac{1-Ds}{C}+1+s\right)},\qquad(A24)$$

$\quad$ where $\Gamma(\ )$ is the Gamma function. The Gamma function has the property $\Gamma(q+1) = q\Gamma(q)$, which allows Eq. (A24) to be
simplified to,

$$Y(s) = \left[\frac{\Gamma(s)}{C}\right]\left[\frac{D\left(\frac{1-Ds}{C}+s\right)\Gamma\left(\frac{1-Ds}{C}\right)}{\Gamma\left(\frac{1-Ds}{C}+1+s\right)} + \frac{(C-D)\left(\frac{1-Ds}{C}\right)\Gamma\left(\frac{1-Ds}{C}\right)}{\Gamma\left(\frac{1-Ds}{C}+1+s\right)}\right],\qquad(A25)$$

and,

$$Y(s) = \left[\frac{\Gamma(s)}{C}\right]\left[\frac{sD\Gamma\left(\frac{1-Ds}{C}\right)+C\left(\frac{1-Ds}{C}\right)\Gamma\left(\frac{1-Ds}{C}\right)}{\Gamma\left(\frac{1-Ds}{C}+1+s\right)}\right] = \left[\frac{\Gamma(s)\Gamma\left(\frac{1-Ds}{C}\right)}{C}\right]\left[\frac{1-Ds+Ds}{\Gamma\left(\frac{1-Ds}{C}+1+s\right)}\right] = \frac{\Gamma(s)\Gamma\left(\frac{1-Ds}{C}\right)}{C\Gamma\left(\frac{1-Ds}{C}+1+s\right)} .\qquad(A26)$$





### A.4 The inverse Mellin transform and solution for $y(z)$

We now take the inverse Mellin transform of Eq. (A26) and solve for $y(z)$ explicitly. The inverse Mellin transform is defined as,

$$y(z) = \frac{1}{2\pi i} \int_{k-\infty i}^{k+\infty i} Y(s) z^{-s} ds \, , \tag{A27}$$

where the integral from $k - \infty i$ to $k + \infty i$ is interpreted as a line integral along a vertical line in the complex plane. For our specific function, the inverse Mellin transform is,

$$y(z) = \frac{1}{2\pi i} \int_{k-\infty i}^{k+\infty i} \frac{z^{-s} \Gamma(s) \Gamma\left(\frac{1-Ds}{C}\right)}{C\Gamma\left(\frac{1-Ds}{C}+1+s\right)} ds \quad where \; 0 < k < \frac{1}{D} \, . \tag{A28}$$

The constraint $0 < k < \frac{1}{D}$ is due to Eq. (A14), the constraint on the real part of $s$ so that Eq. (A2) would converge. This means that the vertical line in the complex plane over which the line integral is taken must fall between $0$ and $\frac{1}{D}$ on the real axis (Fig. A1). We evaluate the integral in Eq. (A28) to find an explicit form of $y(z)$ using the following methodology:

1) Define an appropriate contour in the complex plane to perform a contour integration of $Y(s)z^{-s}$.
2) Use residue integration to evaluate the value of this contour integral.
3) Show that the only part of this contour that does not vanish is the line integral defined in Eq. (A28), meaning the inverse Mellin transform, and therefore $y(z)$, is equal to the value of the contour integral evaluated in step 2.

First, we choose a semicircle contour in the complex plane, with the straight portion as a vertical line crossing the real axis at $s = k$, and the circular portion connecting the ends of this line across the left side of the complex plane (Fig. A1). This contour is consistent with the constraint on the real part of $s$ given in Eq. (A14). We can now define the integral of $Y(s)z^{-s}$ over this contour,

$$\Lambda(z) = \oint Y(s)z^{-s} ds \, . \tag{A29}$$

$\Lambda(z)$ can be expressed as the sum of two line integrals, one over the vertical line portion of the contour, and one over the circular arc portion of the contour,

$$\Lambda(z) = \oint Y(s)z^{-s} ds = \int_{k-Ri}^{k+Ri} Y(s) z^{-s} ds + \int_{arc} Y(s)z^{-s} ds \, , \tag{A30}$$

where $R$ is the radius of the semicircle contour. Allowing $R \to \infty$ leads to,

$$\Lambda(z) = \oint Y(s)z^{-s} ds = \int_{k-\infty i}^{k+\infty i} Y(s) z^{-s} ds + \int_{arc} Y(s)z^{-s} ds \, . \tag{A31}$$

We can now evaluate $\Lambda(z)$ over the infinitely large contour using residue integration. Residue integration relates the value of a contour integral to the sum of the residues of the function being integrated. Residues occur when the function of interest has singularities within the contour. Inspection of $Y(s)z^{-s}$ (i.e. the integrand in Eq. (A26)) inside the semicircle contour shows





that the only component with singularities is $\Gamma(s)$. $\Gamma(s)$ is undefined and has simple poles at $s = -r$ where $r = 0, 1, 2, 3, \ldots \infty$. The residues of the gamma function for each value of $r$ are,

$$\text{Res}(\Gamma(s), -r) = \frac{(-1)^r}{r!} . \tag{A32}$$

380 Using the residue theorem we can evaluate $\Lambda(z)$,

$$\Lambda(z) = \oint Y(s) z^{-s} ds = 2\pi i \sum \text{Res}(Y(s) z^{-s}, -r) = 2\pi i \sum_{r=0}^{\infty} \frac{z^r (-1)^r \Gamma\left(\frac{1+Dr}{C}\right)}{r! C \Gamma\left(\frac{1+Dr}{C}+1-r\right)} , \tag{A33}$$

which contains the sum over the infinite number of residues of $Y(s)z^{-s}$ within the semicircular contour. Substituting this solution into Eq. (A29) gives,

$$\Lambda(z) = \int_{k-\infty i}^{k+\infty i} Y(s) z^{-s} ds + \int_{arc} Y(s) z^{-s} ds = 2\pi i \sum_{r=0}^{\infty} \frac{z^r (-1)^r \Gamma\left(\frac{1+Dr}{C}\right)}{r! C \Gamma\left(\frac{1+Dr}{C}+1-r\right)} . \tag{A34}$$

385 Equation (A34) is the line integral of $Y(s)z^{-s}$ evaluated for the entire contour; now we investigate the contribution of the line integral over just the circular arc portion of the contour. Using the estimation lemma, we can write the following inequality for the line integral over the arc,

$$\left| \int_{arc} Y(s) z^{-s} ds \right| \le ML , \tag{A35}$$

where $L$ is the length of the arc and $M$ is the maximum value of $|Y(s)z^{-s}|$ along the length of the arc. Writing $s$ in terms of its 390 real and imaginary parts, $s = a + bi$, the length of the arc is defined as the product of the central angle and radius of the circle,

$$L = \pi R = \pi \sqrt{a^2 + b^2} . \tag{A36}$$

$|Y(s)z^{-s}|$ can be written as,

$$|Y(s) z^{-s}| = \frac{|z^{-s}||\Gamma(s)|\left|\Gamma\left(\frac{1-Ds}{C}\right)\right|}{\left|C\Gamma\left(\frac{1-Ds}{C}+1+s\right)\right|} . \tag{A37}$$

We are interested in the value of $|Y(s)z^{-s}|$ as $R \to \infty$, therefore, we can approximate the absolute values of the gamma 395 functions using a form of Stirling's formula, valid for values of $s$ as $|b| \to \infty$ ,

$$|\Gamma(s)| = |\Gamma(a+bi)| \sim \sqrt{2\pi} e^{\frac{-\pi|b|}{2}} |b|^{a-\frac{1}{2}} . \tag{A38}$$

Applying Stirling's approximation to Eq. (A37) yields,

$$|Y(s) z^{-s}| \approx \frac{|z^{-a-bi}| \sqrt{2\pi} e^{\frac{-\pi|b|}{2}} |b|^{a-\frac{1}{2}} \sqrt{2\pi} e^{\frac{-\pi\left|-\frac{D}{C}b\right|}{2}} \left|-\frac{D}{C}b\right|^{\frac{1-D}{C}a-\frac{1}{2}}}{C \sqrt{2\pi} e^{\frac{-\pi\left|\left(1-\frac{D}{C}\right)b\right|}{2}} \left|\left(1-\frac{D}{C}\right)b\right|^{\frac{C+1-D}{C}a+1-\frac{1}{2}}} , \tag{A39}$$





which becomes,

$$
|Y(s)z^{-s}| \approx \frac{\sqrt{2\pi}z^{-a}e^{\left[\frac{-\pi}{2}b+\frac{-\pi D}{2}\frac{b}{C}+\frac{\pi}{2}\left(1-\frac{D}{C}\right)b\right]}b^{\left[a-\frac{C+1-D}{C}a+\frac{1-D}{C}a-\frac{3}{2}\right]}\left(\frac{D}{C}\right)^{\left[\frac{1-D}{C}a-\frac{1}{2}\right]}}{c\left(1-\frac{D}{C}\right)^{\left[\frac{C+1-D}{C}a+\frac{1}{2}\right]}} = \frac{\sqrt{\frac{2\pi}{CD-D^2}}z^{-a}e^{\frac{-\pi Db}{C}}b^{-\frac{3}{2}}\left(\frac{D}{C}\right)^{\left[\frac{1-D}{C}a\right]}}{\left(1-\frac{D}{C}\right)^{\left[\frac{C+1-D}{C}a\right]}} ,
\tag{A40}
$$

resulting in,

$$
|Y(s)z^{-s}| \approx \left[\sqrt{\frac{2\pi}{CD-D^2}}\right]\left[\frac{\left(\frac{D}{C}\right)^{\left(\frac{1-D}{C}\right)}}{z\left(1-\frac{D}{C}\right)^{\left(\frac{C+1-D}{C}\right)}}\right]^a\left[e^{\frac{-\pi Db}{C}}\right]\left[b^{-\frac{3}{2}}\right] .
\tag{A41}
$$

Inspecting Eq. (A41), we see that $|Y(s)z^{-s}|$ decreases as $|b|$ increases and $|Y(s)z^{-s}|$ increases as $a$ increases. Therefore, $M$ occurs at the starting and ending point of the arc, where $a = k$. When $a = k$, letting $R \to \infty$ is equivalent to letting $|b| \to \infty$, allowing us to evaluate $ML$,

$$
ML = \lim_{|b|\to\infty} \pi\sqrt{k^2+b^2}\left[\sqrt{\frac{2\pi}{CD-D^2}}\right]\left[\frac{\left(\frac{D}{C}\right)^{\left(\frac{1-D}{C}\right)}}{z\left(1-\frac{D}{C}\right)^{\left(\frac{C+1-D}{C}\right)}}\right]^k\left[e^{\frac{-\pi Db}{C}}\right]\left[b^{-\frac{3}{2}}\right] = 0 .
\tag{A42}
$$

Equation (A42) implies that the contribution of the circular arc portion of the contour vanishes,

$$
0 \leq \left|\int_{arc}Y(s)z^{-s}ds\right| \leq ML = 0 ,
\tag{A43}
$$

which means,

$$
\int_{arc}Y(s)z^{-s}ds = \left|\int_{arc}Y(s)z^{-s}ds\right| = 0 ,
\tag{A44}
$$

and therefore,

$$
y(z) = \frac{1}{2\pi i}\Lambda(z) = \frac{1}{2\pi i}\int_{k-\infty i}^{k+\infty i}Y(s)\,z^{-s}ds = \sum_{r=0}^{\infty}\frac{z^r(-1)^r\Gamma\left(\frac{1+Dr}{C}\right)}{r!C\Gamma\left(\frac{1+Dr}{C}+1-r\right)} ,
\tag{A45}
$$

which is the solution for the general form, Eq. (12).





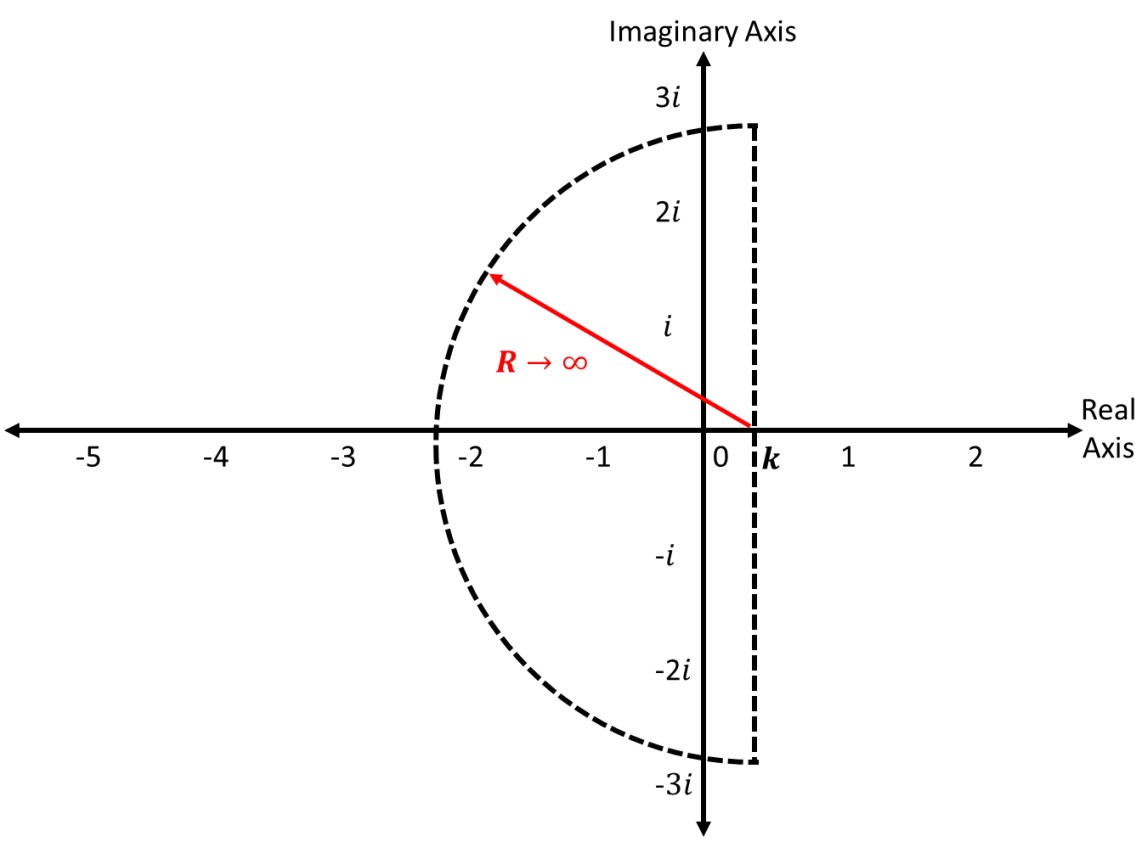

415 **Figure A1: Illustration of the semicircular contour in the complex plane, used to evaluate Eq. (A28). The contour is composed of a vertical line crossing the real axis at $s = k$ and an arc connecting the two ends of the vertical line. The radius of this semicircle is given as $R$. We let $R \to \infty$ so the vertical line portion of the contour will encompass the entire imaginary axis.**

**A.5 Specific functional forms of each parametric Budyko equation**

By comparing the specific forms of the parametric Budyko equations (i.e. Eq. (4), (5), (6), and (7)) to the general form, Eq.

420 (12), we see that $z = -1$ for all of them. We substitute $z = -1$ into Eq. (A45) to produce,

$$y(-1) = \sum_{r=0}^{\infty} \frac{(-1)^{2r}\Gamma\left(\frac{1+Dr}{C}\right)}{r!C\Gamma\left(\frac{1+Dr}{C}+1-r\right)} = \frac{1}{C}\sum_{r=0}^{\infty} \frac{\Gamma\left(\frac{1+Dr}{C}\right)}{\Gamma(r+1)\Gamma\left(\frac{1+Dr}{C}+1-r\right)} . \tag{A46}$$

Substitution of the appropriate expressions for $y$ (i.e. $e^n$ or $e^w$), $C$, and $D$ (Table 1) into Eq. (A46) yields the analytical

solutions for the two parametric forms of the Budyko equation. Specifically these are:

$n$ for arid climates, $\overline{E_0} > \bar{P}$,





425 $\quad n = \ln\left[\dfrac{1}{\ln\left(\frac{\overline{E_0}}{\overline{E}}\right)}\sum_{r=0}^{\infty}\dfrac{\Gamma\left(\frac{1+\ln\left(\frac{\overline{E_0}}{\overline{P}}\right)r}{\ln\left(\frac{\overline{E_0}}{\overline{E}}\right)}\right)}{\Gamma(r+1)\Gamma\left(\frac{1+\ln\left(\frac{\overline{E_0}}{\overline{P}}\right)r}{\ln\left(\frac{\overline{E_0}}{\overline{E}}\right)}+1-r\right)}\right],$ (A47)

$n$ for humid climates, $\overline{E_0} < \bar{P}$,

$n = \ln\left[\dfrac{1}{ln\left(\frac{\bar{P}}{\overline{E}}\right)}\sum_{r=0}^{\infty}\dfrac{\Gamma\left(\frac{1+\ln\left(\frac{\bar{P}}{\overline{E_0}}\right)r}{\ln\left(\frac{\bar{P}}{\overline{E}}\right)}\right)}{\Gamma(r+1)\Gamma\left(\frac{1+ln\left(\frac{\bar{P}}{\overline{E_0}}\right)r}{ln\left(\frac{\bar{P}}{\overline{E}}\right)}+1-r\right)}\right],$ (A48)

$w$ for arid climates, $\overline{E_0} > \bar{P}$,

$w = ln\left[\dfrac{1}{ln\left(\frac{\overline{E_0}+\bar{P}-\overline{E}}{\bar{P}}\right)}\sum_{r=0}^{\infty}\dfrac{\Gamma\left(\frac{1+ln\left(\frac{\overline{E_0}}{\bar{P}}\right)r}{ln\left(\frac{\overline{E_0}+\bar{P}-\overline{E}}{\bar{P}}\right)}\right)}{\Gamma(r+1)\Gamma\left(\frac{1+ln\left(\frac{\overline{E_0}}{\bar{P}}\right)r}{ln\left(\frac{\overline{E_0}+\bar{P}-\overline{E}}{\bar{P}}\right)}+1-r\right)}\right],$ (A49)

430 $\quad w$ for humid climates, $\overline{E_0} < \bar{P}$,

$w = ln\left[\dfrac{1}{ln\left(\frac{\overline{E_0}+\bar{P}-\overline{E}}{\overline{E_0}}\right)}\sum_{r=0}^{\infty}\dfrac{\Gamma\left(\frac{1+ln\left(\frac{\bar{P}}{\overline{E_0}}\right)r}{ln\left(\frac{\overline{E_0}+\bar{P}-\overline{E}}{\overline{E_0}}\right)}\right)}{\Gamma(r+1)\Gamma\left(\frac{1+ln\left(\frac{\bar{P}}{\overline{E_0}}\right)r}{ln\left(\frac{\overline{E_0}+\bar{P}-\overline{E}}{\overline{E_0}}\right)}+1-r\right)}\right].$ (A50)

For critical point catchments (i.e. where $\overline{E_0} = \bar{P}$) the explicit solutions can be found by solving Eq. (4) (or Eq. (6)) and Eq. (5) (or Eq. (7)) algebraically for $n$ and $w$, respectively, yielding,

$n = \dfrac{ln(2)}{ln\left(\frac{\overline{E_0}}{\overline{E}}\right)} = \dfrac{ln(2)}{ln\left(\frac{\bar{P}}{\overline{E}}\right)},$ (A51)

435 $\quad$ and

$w = \dfrac{ln(2)}{ln\left(2-\frac{\overline{E}}{\overline{E_0}}\right)} = \dfrac{ln(2)}{ln\left(2-\frac{\overline{E}}{\bar{P}}\right)}.$ (A52)