# Peer review of "Technical Note: Analytical Inversion of the Parametric Budyko Equations"

_Hydrology and Earth System Sciences, 2020_

## Referee Comment (RC1) · Anonymous Referee #1 · 29 Nov 2020

This manuscript provides a derivation for expressing the Budyko parameters (n or w, more typically also referred to as \omega) explicitly in terms of precipitation, evapo(transpi)ration, and potential ET. The paper argues that this important as past studies could only indirectly infer n or w.

The paper seems technically correct. Being able to explicitly express n and w in terms of precipitation, evapo(transpi)ration, and potential ET can be useful in particular cases, but will not change anything fundamental to the outcome of any study. (Note that the search for factors that determine the catchment-specific parameters of parameterised Budyko curves seems to be largely irrelevant as there is no physical meaning of this parameter that would allow to meaningfully compare this parameter between catchments.)

Since the paper seems technically correct, and some people can use it, I propose to publish this with very minor corrections, but I would encourage the authors to better describe what can and cannot be learned from the catchment specific parameter.

Line 25: To my knowledge, Schreiber (1904) did not use the concept of PET, and this has only been falsely attributed to Schreiber in later publications. It might be worth checking.

L32: Gentine et al. (2012) removed all catchments with Mediterranean and snowy climates, which are (for example in that same dataset) much less accurately following Budyko (see multiple MOPEX studies on climate seasonality effects on E/P and Q/P). Therefore I am not sure it's really appropriate to cite Gentine to support this statement...

---

## Referee Comment (RC2) · Anonymous Referee #2 · 21 Dec 2020

This manuscript is an attempt to investigate the mechanistic understanding of the catchment-specific parameter in parametric Budyko equations. It is an interesting topic in the hydrological studies. However, I have some major concerns on this manuscript. I did not see how this manuscript has made a further step towards explaining the long-term water balance at the catchment scale. By obtaining the mathematical inversion of the catchment-specific parameter from Budyko equations is not helpful in understanding the possible hydrological processes that remain hidden when the non-parametric or parametric Budyko representation is chosen. As mentioned in this paper, what climatic and physiographic features and how they control the long-term water balance are important for explaining the Budyko curve. To achieve this goal, one approach is to express the parameter of parsimonious Budyko equation (e.g., n or $\omega$) in terms of

biophysical features in a way that could be applied to different catchments; the other approach is to explicitly represent the features in the model.

In addition, this paper treats n or $\omega$ as a function of long-term P, E0, and E, and E actually is treated as a function of P, E0. Does it mean that the value of n or $\omega$ is only dependent on the long-term climate? It seems conflict with the existing studies that found the short-term climate variations and catchment features (those could not be explained by the long-term climate) also have impacts on the long-term water balance.

---

## Referee Comment (RC3) · Anonymous Referee #3 · 19 Jan 2021

1) This manuscript analytically derived both forms of the parametric Budyko equations, producing expressions for n and w only in terms of long-term mean P, E0 and E. This derivation is logically sound, but the expressions for n and w are too complex to apply for estimating the values of n and w.

2) This paper lacks rationality analysis of n and w expressions. It is necessary to analyze the rationality of the results using the observation data in some watersheds with different climate and biophysical features.

3) The expressions derived in this paper are only suitable for natural watersheds or for the analysis of climate-vegetation-hydrology equilibrium. However, it can not be used to estimate the impact of land use changes or vegetation changes due to human activities on the water balance of watersheds.

---

## Referee Comment (RC4) · Anonymous Referee #4 · 3 Feb 2021

This technical note outlines the steps for analytically deriving the values of the parameters used in the two main versions of the Budyko model; w for the Fu version and n for the Choudhury-Yang version. The authors derive these as functions of the main model variables: average E, P and Ep. Their key point is that the w/n parameters are only correctly derived as functions of three model variables and should not be expected to be physically related to any other biophysical process, except through the dependence other processes have on E, Ep and P themselves. They conclude that this derivation provides a needed solution to the question of how w/n are related to biophysical processes.

I can not comment on the maths in this article as it is beyond my expertise. I can comment on the concepts and implications of the maths. I have two main points to

make. Overall, I find nothing technically wrong with the article but question the value of its contribution.

Firstly, the derivation of analytical solutions to the two parameters doesn't seem like a new contribution. Sposito [2017, Understanding the Budyko Equation, Water, 9(40)] has done similar including analytically deriving the data space of each parameter. I seek guidance from the editor on the requirements for novelty in technical notes and ask the authors to explain how their study provides an advance over what has already been done. My other point is that, while the authors provide an analytical solution to how to define w and n, I am unsure of what problem it is that they are solving. They indicate the problem has been stated in the literature (L62): "Notably, there has not been an analytical derivation illustrating how ðİŚŻ and ðİŚď relate to biophysical features, though the importance of doing so has been noted many times." and (L169): "...the literature-identified need of an analytical expression." However, the motivation behind trying to predict a catchment's parameter value is to be able to use Budyko to make predictions about E and Q based on Ep and P – that is, in ungauged catchments. The analytical definitions of w and n are given here as functions of E and so cannot address the need most of the literature is trying to address, which is prediction in ungauged catchments. Further, the need stated in the literature calls for a biophysical understanding of the parameters. The authors claim their solution "....thus fulfills the literature-identified need of an analytical expression for ðİŚŻ and ðİŚď in terms of biophysical features." . While they have provided an analytical solution I am not convinced it provides a solution that is any more connected to biophysical features any more than the original formulations are. Another way of saying this is, the solution doesn't provide any greater biophysical understanding of the meaning of w or n than previously existed, nor does it make Budyko any more useful. Again, how does this then address the need to be able to predict w and n ungauged catchments? I think that the authors need to rethink what is the question they are trying to address and ensure it represents an advance in the use of Budyko to make hydrological predictions.

---

## Author Comment (AC1) · 4 Mar 2021

We thank the reviewer for these helpful comments. Reviewer comments are listed below, along with our response to each. In some cases, we describe the proposed revisions to the manuscript (with line numbers), but we recognize that the revised manuscript is requested in a subsequent step.

**Comment 1:**
**This manuscript provides a derivation for expressing the Budyko parameters (n or w, more typically also referred to as omega) explicitly in terms of precipitation, evapo(transpi)ration, and potential ET. The paper argues that this important as past studies could only indirectly infer n or w.**

Response 1:
This is an accurate representation, though we note that another outcome of this work is to illustrate how $n$ and $w$ depend on biophysical features specifically through the dependence of $\overline{E_0}$, $\bar{P}$, and $\bar{E}$ on those same features (e.g., lines 15, 60-63, and 165-170).

**Comment 2:**
**The paper seems technically correct. Being able to explicitly express n and w in terms of precipitation, evapo(transpi)ration, and potential ET can be useful in particular cases, but will not change anything fundamental to the outcome of any study. (Note that the search for factors that determine the catchment-specific parameters of parameterized Budyko curves seems to be largely irrelevant as there is no physical meaning of this parameter that would allow to meaningfully compare this parameter between catchments.)**

Response 2:
We thank the reviewer for supporting the technical aspects of the derivation. We wholeheartedly agree with the reviewer on both points, which we address in much more detail in the companion research article to the technical note (Reaver et al., 2020) (hess-2020-584, "Reinterpreting the Budyko Framework", cited on page 3, line 57). We note that the primary aim of this technical note is to analytically invert the parametric Budyko equations and verify that the resulting explicit expressions are correct. A secondary aim is to improve the mechanistic understanding of $n$ and $w$, however, this theme is more completely developed in the companion article (Reaver et al. 2020).

Given the relatively narrow scope of the technical note, we thus focus on the technical details of the analytical inversion. However, in our response to Comment 3 (below), we do propose several revisions to the abstract, introduction, and discussion sections that aim to better motivate the study and more clearly describe how explicit expressions for $n$ and $w$ show the direct dependence of $n$ and $w$ on $\bar{P}$, $\overline{E_0}$, and $\bar{E}$, thus illustrating their lack of physical meaning.

**Comment 3:**
**Since the paper seems technically correct, and some people can use it, I propose to publish this with very minor corrections, but I would encourage the authors to better describe what can and cannot be learned from the catchment specific parameter.**

Response 3:
We thank the reviewer for the recommendation to publish with very minor corrections and agree that it would be useful to better describe what can and cannot be learned from $n$ and $w$. We propose to expand the explanation of our motivation and interpretation with the following edits:

1) Revise the abstract to highlight the direct dependence of dependence of $n$ and $w$ on $\bar{P}$, $\overline{E_0}$, and $\bar{E}$ and make the point that, for practical applications (e.g., hydrological predictions), the parametric Budyko equations lack utility:

*"The non-parametric Budyko framework provides empirical relationships between a catchment's long-term mean evapotranspiration ($\bar{E}$) and the aridity index, defined as the ratio of mean rainfall depth ($\bar{P}$) to mean potential evapotranspiration ($\overline{E_0}$). The parametric Budyko equations attempt to generalize this framework by introducing a catchment-specific parameter ($n$ or $w$), intended to represent differences in catchment climate and landscape features. Many studies have developed complex regression relationships for the catchment-specific parameter in terms of biophysical features, all of which use known values of $\bar{P}$, $\overline{E_0}$, and $\bar{E}$ to numerically invert the parametric Budyko equations to obtain values of $n$ or $w$. Critically, the introduction of $n$ or $w$ renders the parametric Budyko equations underdetermined, precluding their use in predicting $\bar{E}$ and severely limiting their practical application. In this study, we analytically invert both forms of the parametric Budyko equations, producing expressions for $n$ and $w$ only in terms of $\bar{P}$, $\overline{E_0}$, and $\bar{E}$. These expressions allow for $n$ and $w$ to be explicitly expressed in terms of biophysical features through the dependence of $\bar{P}$, $\overline{E_0}$, and $\bar{E}$ on those same features, illustrating explicitly why the parametric Budyko equations cannot be used for predicting $\bar{E}$."*

2) Revise the introduction (pages 2-3, lines 44-66) to better contextualize the note within a broader fundamental and conceptual critique of the parametric Budyko framework and catchment-specific parameters:

*"The catchment-specific parameter ($n$ or $w$) has been described as an empirical effective parameter representing the influence of all catchment biophysical features, other than $\bar{P}$ and $\overline{E_0}$, on $\bar{E}$ (Wang et al., 2016a), though this interpretation does not arise from the derivations of Eq. (4) and (6) (Yang et al., 2008) or Eq. (5) and (7) (Zhang et al., 2004). Additionally, the functional forms of the parametric Budyko equations have typically been interpreted as representing the evaporative behavior of individual catchments under different aridity indices (e.g., (Roderick and Farquhar, 2011; Wang and Hejazi, 2011; Yang and Yang, 2011; Wang et al., 2016b; Zhou et al., 2016; Shen et al., 2017; Zhang et al., 2016; Milly et al., 2018)), though this interpretation had not been justified experimentally or observationally. To the contrary, empirical tests of this interpretation strongly suggest that the parametric Budyko equations do not describe the long-term evaporative behavior of catchments, which implies that they are not physically meaningful (Reaver et al., 2020). This means that the values of $n$ and $w$ are not transferable between catchments or across time for individual catchments and thus cannot be related to physical properties in the same manner as the effective parameters in other well-accepted empirical hydrological relationships (e.g., the roughness coefficient in Manning's equation and hydraulic conductivity in Darcy's Law). This renders the parametric Budyko equations under-determined for predicting $\bar{E}$ from only $\bar{P}$ and $\overline{E_0}$ (i.e., one equation with two unknowns, $\bar{E}$ and $n$ or $w$). The non-transferability and under-determined nature of these equations has been implicitly acknowledged previously in the literature, e.g., (Zhang et al., 2004; Wang et al., 2016a; Greve et al., 2015), where it has been noted that it is not possible to obtain the value of $n$ or $w$ for a specific catchment a priori; one must first estimate $\bar{E}$, $\bar{P}$ and $\overline{E_0}$ and then invert either Eq. (4), (5), (6), or (7). This lack of predictive ability effectively precludes the practical application of these equations.*

*Despite this fact, many studies have adopted the "biophysical features" interpretation of $n$ and $w$ and have developed complex regression relationships for the catchment-specific parameter in terms of various*

*climate and landscape features (Yang et al., 2007;Donohue et al., 2012;Yang et al., 2009;Shao et al., 2012;Li et al., 2013;Xu et al., 2013;Cong et al., 2015;Yang et al., 2016;Zhang et al., 2018;Abatzoglou and Ficklin, 2017;Xing et al., 2018;Zhao et al., 2020;Ning et al., 2020b;Ning et al., 2020a;Li et al., 2020b;Li et al., 2020a;Zhang et al., 2019;Ning et al., 2019;Bai et al., 2019;Ning et al., 2017). In all such studies, known values of $\bar{P}$, $\overline{E_0}$, and $\bar{E}$, estimated empirically or via modelling, are used to numerically invert the parametric Budyko equations to obtain values of the catchment-specific parameter, which are then regressed against various biophysical features. The expressions obtained from such endeavours vary significantly between studies, both in their functional forms and what biophysical features are included in the regression, making it difficult to develop a consistent mechanistic understanding of the catchment-specific parameter (Reaver et al., 2020). The intention of these regression expressions is often to substitute them in Eq. (4), (5), (6), or (7) in order to predict $\bar{E}$. This is a circular process, where an estimate of $\bar{E}$ was used to estimate $n$ or $w$, which is then used to estimate $\bar{E}$. In practice, since the inclusion of the parametric Budyko framework adds no new information (Reaver et al., 2020), $n$ and $w$ could be eliminated from this process by fitting the regression models to the already estimated values of $\bar{E}$ directly, bypassing the parametric Budyko framework altogether.*

*Given the large number of studies seeking to relate the catchment-specific parameter to biophysical features, it seems that the process of numerically inverting the parametric Budyko equations, coupled with the assumption that they are empirically valid and physically meaningful, obscures their under-determined nature and the complete dependence of $n$ and $w$ on $\bar{P}$, $\overline{E_0}$, and $\bar{E}$. In this study, we analytically invert both forms of the parametric Budyko equations. The resulting expressions give $n$ and $w$ only in terms of $\bar{P}$, $\overline{E_0}$, and $\bar{E}$, illustrating that if $n$ and $w$ depend on any biophysical features, it is due directly to the dependence of $\bar{P}$, $\overline{E_0}$, or $\bar{E}$ on those same features. Notably, there has not been an analytical derivation illustrating how $n$ and $w$ relate to biophysical features, though the importance of doing so has been noted many times (Zhang et al., 2004;Yang et al., 2008;Donohue et al., 2012;Xu et al., 2013;Greve et al., 2015;Wang et al., 2016a;Zhang et al., 2018). The expressions we develop here for $n$ and $w$ satisfy this need, providing a general expression for the dependence of $n$ and $w$ on any possible biophysical features through the dependence of $\bar{P}$, $\overline{E_0}$, and $\bar{E}$ on those same features."*

3) Revise the Discussion and Conclusions section (pages 9, lines 165-170) to summarize interpretations regarding the utility of catchment-specific parameters and the overall parametric approach:

*"Notably, the explicit analytical expression for $n$ and $w$ from Eq. (14) illustrates that the value of the catchment-specific parameter is only determined by $\bar{E}$, $\overline{E_0}$, and $\bar{P}$. Therefore, if $n$ or $w$ depend on biophysical features, it is directly due to the dependence of $\bar{E}$, $\overline{E_0}$, or $\bar{P}$ on those features. In short, this means that Eq. (14) is the general solution for how $n$ and $w$ depend on biophysical features. By substituting $\bar{E}$, $\overline{E_0}$, or $\bar{P}$ as functions of specific biophysical features into Eq. (14), one obtains the expression for $n$ and $w$ as a function of those features. Eq. (14) thus fulfills the literature-identified need of an analytical expression for $n$ and $w$ in terms of biophysical features. The main implication of the direct dependence of $n$ and $w$ on $\bar{E}$, $\overline{E_0}$, and $\bar{P}$ is that $\bar{E}$, $\overline{E_0}$, and $\bar{P}$ must always be estimated prior to obtaining a value of $n$ or $w$, meaning the parametric Budyko equations are unable to independently predict $\bar{E}$ from $\overline{E_0}$ and $\bar{P}$. Since the prediction of $\bar{E}$ and its possible temporal evolution are the primary applications of the Budyko framework, the practical utilities of the parametric Budyko equations are severely limited."*

**Comment 4:**
**Line 25: To my knowledge, Schreiber (1904) did not use the concept of PET, and this has only been falsely attributed to Schreiber in later publications. It might be worth checking.**

Response 4:
We thank the reviewer for calling this to our attention. We revisited the text of Schreiber (1904) to assess whether the concept of potential evapotranspiration was utilized within the manuscript. While the concept of potential evapotranspiration is not explicitly stated, Schreiber (1904) has a functionally equivalent constant "k" in its place. He refers to "k" as the limiting value that the difference between mean annual precipitation and runoff ($\bar{P} - \bar{Q}$, referred to as "die Rückstandshöhe" or the catchment's residue/hold-back height) approaches as precipitation becomes large (i.e., $\bar{P} \to \infty$). Quoting the specific passage:

*Je größer x [der jährichen Niederschlagschöhe] wird, um so kleiner wird $\frac{k}{x}$, so daß man für sehr große x*

$$y = x - k$$

*[die jähriche Abflußhöhe] setzen kann. Heiraus ergibt sich sofort die physikalische Bedeutung des Exponenten k als die Größe, der sich die Differenz zwischen Niederschlag und Abfluß [y] um so mehr nähert, je größer der Niederschlag selbst wird. Dieses Verhältnis scheint mir in der Natur des Problemes begründet zu sein. Die Differenz*

$$z = x - y$$

*kann man als die Rückstandshöhe bezeichen.* Schreiber (1904), page 3.

In our current language, the constant k would be the mean annual value of evapotranspiration under energy-limited conditions, i.e., the mean annual potential evapotranspiration, $\overline{E_0}$. However, while constant k is functionally equivalent to $\overline{E_0}$, Schreiber (1904) does not discuss or specify how the water that does not become discharge is being "held back" (i.e., does not discuss it as being evaporated) and therefore does not explicitly introduce the concept of potential evapotranspiration. We propose the following edits to the manuscript to more accurately reflect the contribution of Schreiber (1904):

1) Add the following sentences immediately following Eq. (3) to clarify Schreiber's contribution:

*"However, we note that Eq. (2) was originally introduced by Schreiber (1904) with a constant "K" in place of $\overline{E_0}$. While "K" was functionally equivalent to $\overline{E_0}$ in its implementation, its physical interpretation in relation to catchment hydrology was only partially developed by Schreiber (1904). Subsequent investigations by Ol'Dekop (1911) ascribed the concept of maximum possible evaporation (i.e., potential evapotranspiration) to "K", as detailed in Andréassian et al. (2016)."*

2) Modify "*Equations (2) and (3) were selected...*" (page 2, lines 34-36) to "T*he functional forms of Eq. (2) and (3) were selected...*".

**Comment 5:**
**L32: Gentine et al. (2012) removed all catchments with Mediterranean and snowy climates, which are (for example in that same dataset) much less accurately following Budyko (see multiple MOPEX studies**

**on climate seasonality effects on E/P and Q/P). Therefore I am not sure it's really appropriate to cite Gentine to support this statement…**

Response 5:

We thank the reviewer for calling this to our attention. Gentine et al. (2012) describe their methodology for excluding catchments as follows:

*"We exclude those basins with missing data; records shorter than 50 years; significant topographical gradients, i.e., elevation changes greater than 1000m or slope steeper than 15 percent, since these basins are likely to span distinct climate regimes and associated impacts on the hydrologic cycle; important anthropogenic modifications (e.g., irrigation, reservoirs) based on the estimates of Wang and Hejazi [2011]. A total of 77 [out of 431] basins was removed from the analysis.*" Gentine et al. (2012), page 2.

Thus, from our reading, Gentine et al. (2012) do not exclude Mediterranean (i.e., hot dry summers and cool wet winters) nor snowy climates. In fact, several catchments with Mediterranean climates were specifically included in the analysis, labeled as "Out-of-phase climates" by Gentine et al. (2012), and the authors specifically note that phase differences did not cause significant departures from the non-parametric Budyko curve:

*"The seasonality between rainfall and potential evaporation does not alter the fit of the basins to the Budyko curve. Noticeably, no systematic biases are present for summer or winter-dominated rainfall regimes (Figure 1b)."* Gentine et al. (2012), pages 2-3.

Even though Mediterranean and perennial snowy climates were not explicitly excluded from Gentine et al. (2012), we acknowledge the reviewer's point that catchments with such climates may be underrepresented in the sample—and therefore the ~10% error referenced by Gentine et al. (2012) may be too low. As a check, we computed the distribution of absolute percent errors from the non-parametric Budyko curve for all MOPEX catchments (Schaake et al., 2006) with sufficient data to calculate $\overline{E_0}$, $\overline{P}$, and $\overline{E}$ (428 out 438 total catchments). Plotting these catchments in Budyko space (Figure 1), we see more spread around the Budyko curve than found in Gentine et al. (2012), however the mean error is 10.3% (Figure 2), closely aligned with both Gentine et al. (2012) and Budyko and Zubenok (1961).

We thus chose to retain the Gentine et al. (2012) reference, but propose the following edits (page 2, lines 31-32) to acknowledge that the 10% uncertainty number refers to the mean of the error distribution:

*"The geometric mean of Eq. (2) and (3) have been shown to predict $\frac{\overline{E}}{\overline{P}}$ with a mean uncertainty of ~10% (Budyko and Zubenok, 1961;Gentine et al., 2012) for ungauged basins if $\overline{P}$ and $\overline{E_0}$ are known."*

[Figure]

Figure 1: 428 MOPEX catchments (red dots) compared to the non-parametric Budyko equation (blue curve) plotted in Budyko space.

[Figure]

Figure 2: Histogram of the absolute percent errors between the 428 MOPEX catchments and the non-parametric Budyko equation. The mean percent error (vertical dashed red line) is 10.3%.

**Refrences:**

Abatzoglou, J. T., and Ficklin, D. L.: Climatic and physiographic controls of spatial variability in surface water balance over the contiguous United States using the Budyko relationship, Water Resources Research, 53, 7630-7643, 2017.

Andréassian, V., Mander, Ü., and Pae, T.: The Budyko hypothesis before Budyko: The hydrological legacy of Evald Oldekop, Journal of Hydrology, 535, 386-391, 10.1016/j.jhydrol.2016.02.002, 2016.

[revised manuscript text omitted]

---

## Author Comment (AC2) · 4 Mar 2021

We thank the reviewer for these helpful comments. Reviewer comments are listed below, along with our response to each. In some cases, we describe the proposed revisions to the manuscript (with line numbers) or refer to proposed revisions described in our responses to the other reviewers, but we recognize that the revised manuscript is requested in a subsequent step.

**Comment 1:**
**This manuscript is an attempt to investigate the mechanistic understanding of the catchment-specific parameter in parametric Budyko equations. It is an interesting topic in the hydrological studies.**

Response 1:
We appreciate the reviewer's interest in our manuscript but stress that the primary aim of this technical note is to present the technical details of analytically inverting the parametric Budyko equations. We strongly agree that improving our mechanistic understanding of the catchment-specific parameter is important and address this concept in several places in the text (page 1, lines 15-17; page 3, lines 55-62; and page 9, lines 165-168), but we believe that the technical aspects of the analytical inversion and resulting explicit expressions for $n$ and $w$ are novel additions to the literature on their own. We note that the companion research article to this technical note, (Reaver et al., 2020) (hess-2020-584, Reinterpreting the Budyko Framework, referenced on page 3, line 57 of the technical note), provides a much more in-depth analysis and critique of the catchment-specific parameter and its use in the parametric Budyko framework.

**Comment 2:**
**However, I have some major concerns on this manuscript. I did not see how this manuscript has made a further step towards explaining the longterm water balance at the catchment scale. By obtaining the mathematical inversion of the catchment-specific parameter from Budyko equations is not helpful in understanding the possible hydrological processes that remain hidden when the non-parametric or parametric Budyko representation is chosen.**

Response 2:
As noted above, the primary objective of this technical note is the analytical inversion of the parametric Budyko equations and resulting explicit expressions for $n$ and $w$ (not explaining the long-term catchment water balance). Regarding the utility of the inversion, we illustrate that the dependence of $n$ and $w$ on any "hidden" biophysical processes must be through the dependence of $\bar{P}$, $\overline{E_0}$, and $\bar{E}$ on those same features and processes, thus advancing our understanding of the parametric Budyko equations (and limitations to their utility). While the companion research article (Reaver et al., 2020) explicitly states that the parametric Budyko equations cannot be productively used to understand the long-term water balance (nor to identify potential "hidden" hydrological processes), this point could be more clearly articulated in this technical note. We thus propose revisions to the abstract, introduction, and discussion and conclusions sections to better motivate the study and summarize our interpretations regarding the utility of catchment-specific parameters and the overall parametric approach. These proposed revisions are given in our response to **Reviewer 1, Comment 3.**

**Comment 3:**
**As mentioned in this paper, what climatic and physiographic features and how they control the long-term water balance are important for explaining the Budyko curve. To achieve this goal, one approach is to express the parameter of parsimonious Budyko equation (e.g., n or ω) in terms of biophysical**

**features in a way that could be applied to different catchments; the other approach is to explicitly represent the features in the model.**

Response 3:
The question of why the Budyko curve emerges from the aggregate behavior of many catchments across a wide range of aridity indices is not the topic of this manuscript. Instead, we are focused on inverting the parametric Budyko equations, while secondarily commenting on how the derived expressions for *n* and *w* provide a general relationship between the catchment-specific parameters and biophysical features through the dependence of $\overline{E_0}$, $\bar{P}$, and $\bar{E}$ on those same features. As we emphasized in our response to **Reviewer 2, Comment 2** above, understanding how *n* and *w* depend on biophysical features is tangential to understanding or explaining the Budyko curve. The reviewer also suggests that *n* or *w* should be expressed in terms of biophysical features in a way that can be applied to other catchments. Our explicit expressions (Equation 14, page 4, line 85) do exactly that through the direct dependence of $\overline{E_0}$, $\bar{P}$, and $\bar{E}$ on biophysical features. However, we again note that, if the dependence of $\overline{E_0}$, $\bar{P}$, and $\bar{E}$ on biophysical features are known (which they must be in order to calculate n or w), then the long-term water balance could be understood directly (the reviewer's suggested second approach) without the need for the parametric Budyko framework. Both of these topics are explored in detail in the companion research article to this technical note (Reaver et al., 2020) (hess-2020-584, Reinterpreting the Budyko Framework, cited on page 3, line 57).

**Comment 4:**
**In addition, this paper treats n or ω as a function of long-term P, E0, and E, and E actually is treated as a function of P, E0. Does it mean that the value of n or ω is only dependent on the long-term climate? It seems conflict with the existing studies that found the short-term climate variations and catchment features (those could not be explained by the long-term climate) also have impacts on the long-term water balance.**

Response 4:
The reviewer is correct that we derive *n* and *w* solely as functions of long-term of $\overline{E_0}$, $\bar{P}$, and $\bar{E}$. However, it is not correct that we treat $\bar{E}$ only as a function of $\overline{E_0}$ and $\bar{P}$. This confusion may be due to the introduction of Equation 1 (page 1, line 24), which states that non-parametric Budyko equations are functions of the aridity index (i.e., functions of only $\overline{E_0}$ and $\bar{P}$). Critically, this only applies to the non-parametric Budyko equations. When the parametric Budyko equations were formally derived (Yang et al., 2008;Zhang et al., 2004), the starting point of the derivation was the relaxation of the requirement that $\bar{E} = f_0(\overline{E_0}, \bar{P})$ to the implicit relationship $\bar{E} = f_1(\overline{E_0}, \bar{P}, \bar{E})$ (i.e., $\bar{E}$ depends on $\bar{E}$). In both derivations, this relaxation eventually leads to the introduction of an arbitrary constant, which eventually became *n* and *w* in the parametric Budyko equations. By inspecting the parametric Budyko equations (Equations 4, 5, 6, and 7, page 2 , lines 36-43), it is clear that *n* and *w* are functions of $\overline{E_0}$, $\bar{P}$, and $\bar{E}$; our technical note only gives the explicit form of those functions (Equation 14, page 4, line 85).

In short, *n* and *w* are only dependent on $\overline{E_0}$, $\bar{P}$, and $\bar{E}$. Short-term climate variations and catchment features are likely to impact $\bar{E}$ (i.e., the long term water balance) and would also impact *n* and *w* through their dependence on $\bar{E}$. Therefore, our results are not in conflict with existing studies that have found catchment properties other than $\overline{E_0}$ and $\bar{P}$ to impact the long term water balance. To make this clearer we propose to revise our description of the parametric Budyko equation development (page 2, lines 33-35) to explicitly mention its implicit nature:

*"In an attempt to generalize the non-parametric Budyko framework and explain deviations in $\frac{\bar{E}}{\bar{P}}$ from the central tendency of the empirically observed catchment clustering pattern, the implicit relationship, $\bar{E} = f_1(\overline{E_0}, \bar{P}, \bar{E})$, was proposed, resulting in two parametric Budyko equations (Turc, 1953; Choudhury, 1999; Mezentsev, 1955; Yang et al., 2008),…"*

**References:**

Reaver, N. G. F., Kaplan, D. A., Klammler, H., and Jawitz, J. W.: Reinterpreting the Budyko Framework, Hydrol. Earth Syst. Sci. Discuss., 2020, 1-31, 10.5194/hess-2020-584, 2020.

Yang, H., Yang, D., Lei, Z., and Sun, F.: New analytical derivation of the mean annual water-energy balance equation, Water Resources Research, 44, n/a-n/a, 10.1029/2007wr006135, 2008.

Zhang, L., Hickel, K., Dawes, W. R., Chiew, F. H. S., Western, A. W., and Briggs, P. R.: A rational function approach for estimating mean annual evapotranspiration, Water Resources Research, 40, n/a-n/a, 10.1029/2003wr002710, 2004.

---

## Author Comment (AC3) · 4 Mar 2021

We thank the reviewer for these helpful comments. Reviewer comments are listed below, along with our response to each. In some cases, we describe the proposed revisions to the manuscript (with line numbers) or refer to proposed revisions described in our responses to the other reviewers, but we recognize that the revised manuscript is requested in a subsequent step.

**Comment 1:**
**This manuscript analytically derived both forms of the parametric Budyko equations, producing expressions for n and w only in terms of long-term mean P, E0 and E.**

Response 1:
To clarify, our manuscript does not derive both forms of the parametric Budyko equations. Rather, we analytically inverted the existing two parametric Budyko equations, producing expressions for *n* and *w* in terms of precipitation, evapotranspiration, and potential evapotranspiration.

**Comment 2:**
**This derivation is logically sound, but the expressions for n and w are too complex to apply for estimating the values of n and w.**

Response 2:
We agree that the full solutions are complex, however, in the context of advancing our understanding of *n* and *w*, the degree of their complexity is largely irrelevant since they are the only extant explicit expressions for *n* and *w*. Therefore, they are currently the only way to estimate *n* and *w* explicitly as well as being the only way to directly relate *n* and *w* to biophysical features through $\bar{P}$, $\overline{E_0}$, and $\bar{E}$. Prior to the development of these explicit expressions, *n* and *w* could only be determined from $\bar{P}$, $\overline{E_0}$, and $\bar{E}$ by numerically solving the parametric Budyko equations. It is plausible that the expressions presented could be simplified to a "less complex" form or that a simpler analytical inversion could be developed, however, until such events occur, these expressions are the simplest (and only) explicit expressions for *n* and *w*.

We acknowledge the reviewer's point that when directly computing *n* and *w* from values of $\bar{P}$, $\overline{E_0}$, and $\bar{E}$, the explicit expressions we derived do not offer a significant advantage over the standard numerical inversion method, since in practice both methods must implemented with computational algorithms. However, the direct computation of *n* and *w* is not the primary utility of the explicit expressions. Rather, they illustrate that the dependence of *n* and *w* on any "hidden" biophysical processes must be through the dependence of $\bar{P}$, $\overline{E_0}$, and $\bar{E}$ on those same features and processes, thus advancing our understanding of the parametric Budyko equations (as well as limitations to their utility). The direct dependence of *n* and *w* on $\bar{P}$, $\overline{E_0}$, and $\bar{E}$ is explored in detail in the companion research article (Reaver et al., 2020) (hess-2020-584, Reinterpreting the Budyko Framework, cited on page 3, line 57), however, this point could be more clearly articulated in this technical note. We thus propose revisions to the abstract, introduction, and discussion and conclusions sections to better motivate the study and summarize our interpretations regarding the utility of catchment-specific parameters and the overall parametric approach. These proposed revisions are given in our response to **Reviewer 1, Comment 3.**

**Comment 3:**
**This paper lacks rationality analysis of n and w expressions. It is necessary to analyze the rationality of the results using the observation data in some watersheds with different climate and biophysical features.**

Response 3:

It is unclear what the reviewer is referring to when requesting a "rationality analysis", nor why it is necessary or relevant to this technical note. As we note in the manuscript, the primary aim of this technical note was to analytically invert the parametric Budyko equations and verify that the derived solutions were valid by comparison to numerical inversion. Thus, it is not clear how observational data from real catchments would be incorporated into the current scope of the manuscript. While it is possible to take observational data (e.g., $\bar{P}$, $\overline{E_0}$, and $\bar{E}$) from different catchments to estimate $n$ or $w$ (using Equation 14) and compare them with the numerical solution of the implicit parametric Budyko equations (Equations 4, 5, 6, and 7), the same approach can be applied to any selected values of $\bar{P}$, $\overline{E_0}$, and $\bar{E}_0$ (as was done in the manuscript; Figures 1-4, pages 6-8, lines 124-155). Importantly, the source of the values for $\bar{P}$, $\overline{E_0}$, and $\bar{E}$ (e.g., computed from observational data or chosen) is not important for verifying the explicit expressions

**Comment 4:**

**The expressions derived in this paper are only suitable for natural watersheds or for the analysis of climate-vegetation-hydrology equilibrium. However, it can not be used to estimate the impact of land use changes or vegetation changes due to human activities on the water balance of watersheds.**

Response 4:

The assertion that the derived expressions for $n$ and $w$ are only suitable for natural watersheds or for the analysis of climate-vegetation-hydrology equilibrium is both unsubstantiated and incorrect. Equation 14 (page 4, line 85) allows one to explicitly estimate the value of $n$ or $w$ for set of $\bar{P}$, $\overline{E_0}$, and $\bar{E}$ values no matter what the source of those values is (e.g., human-impacted watershed, chosen at random, natural watershed, etc.). We agree that the explicit expression for $n$ and $w$ cannot be used to estimate the impact of anthropogenic land use change on the catchment water balance, however this was not the goal of the manuscript, nor a claim that we made. Rather, we stated that Equation 14 (page 4, line 85) provides the "general expression for the dependence of $n$ and $w$ on any possible biophysical features through the dependence of $\bar{P}$, $\overline{E_0}$, and $\bar{E}$ on those same features" (page 3, lines 65-66). In the companion research article (Reaver et al., 2020), we specifically state that the parametric Budyko framework generally cannot be used to estimate the impacts of anthropogenic land use change on the catchment water balance.

**References:**

[revised manuscript text omitted]

---

## Author Comment (AC4) · 4 Mar 2021

We thank the reviewer for these helpful comments. Reviewer comments are listed below, along with our response to each. In some cases, we describe the proposed revisions to the manuscript (with line numbers) or refer to proposed revisions described in our responses to the other reviewers, but we recognize that the revised manuscript is requested in a subsequent step.

**Comment 1:**
**This technical note outlines the steps for analytically deriving the values of the parameters used in the two main versions of the Budyko model; w for the Fu version and n for the Choudhury-Yang version. The authors derive these as functions of the main model variables: average E, P and Ep. Their key point is that the w/n parameters are only correctly derived as functions of three model variables and should not be expected to be physically related to any other biophysical process, except through the dependence other processes have on E, Ep and P themselves. They conclude that this derivation provides a needed solution to the question of how w/n are related to biophysical processes.**

Response 1:
We thank the reviewer for the comprehensive and accurate summary.

**Comment 2:**
**I can not comment on the maths in this article as it is beyond my expertise. I can comment on the concepts and implications of the maths. I have two main points to make. Overall, I find nothing technically wrong with the article but question the value of its contribution.**

Response 2:
As noted in our responses to other reviewers, the primary aim of our technical note was to present the technical details of analytically inverting the parametric Budyko equations, with a secondary aim of providing a framework for improved mechanistic understanding of *n* and *w*. We aim to better support the value of the contribution, particularly focusing on the context and interpretation of the second aim, both in our proposed edits (see response to **Reviewer 1, Comment 3**) and in responses to the reviewer's subsequent comments.

**Comment 3:**
**Firstly, the derivation of analytical solutions to the two parameters doesn't seem like a new contribution. Sposito [2017, Understanding the Budyko Equation, Water, 9(40)] has done similar including analytically deriving the data space of each parameter. I seek guidance from the editor on the requirements for novelty in technical notes and ask the authors to explain how their study provides an advance over what has already been done.**

Response 3:
We thank the reviewer for connecting our work and that of Sposito (2017). Indeed, we were aware of Sposito (2017), and his work is cited in the companion research article to this technical note (Reaver et al., 2020) (hess-2020-584, Reinterpreting the Budyko Framework, cited on page 3, line 57). However, Sposito (2017) does not analytically invert the parametric Budyko equations as we did in this technical note, nor are we aware of any other study which does so. Since no other known study has produced the analytical inversion of either parametric Budyko equation, we can say with a high degree of certainty that these results are novel. Additionally, Sposito (2017) does not analytically derive the data space of each parameter (i.e., $n \in (0, \infty)$ and $w \in (1, \infty)$) as suggested by the reviewer, rather he reports the range of values that $n$ and $w$ can take that were determined in the formal derivations of the parametric Budyko

equations (Yang et al., 2008;Zhang et al., 2004;Fu, 1981). We also report the values that $n$ and $w$ can take in our technical note (page 5, lines 112-114) and confirm that the explicit expressions for $n$ and $w$ (Equation 15, page 4, line 89) have the same data space (pages 5-6, lines 114-123).

The primary aim of Sposito (2017) was to show that if one postulates that the function, $f(\ )$, of the "original" Budyko hypothesis, $\bar{E} = f(\bar{P}, \overline{E_0})$, is homogeneous, then by borrowing concepts from equilibrium thermodynamics, one can derive several extant non-parametric and parametric Budyko equations (specifically, Equations 2, 4, and 5, in our technical note) from their Legendre transformations. Sposito (2017) shows that if one explicitly defines how $\bar{E}$ changes in response to changes in both $\bar{P}$ and $\overline{E_0}$ (i.e., the partial derivatives, $\left(\frac{\partial \bar{E}}{\partial \bar{P}}\right)_{\overline{E_0}}$ and $\left(\frac{\partial \bar{E}}{\partial \overline{E_0}}\right)_{\bar{P}}$), those relationships can be used to define the Legendre transformation, which can be used derived the resulting Budyko curve. Furthermore, Sposito (2017) critically reviews the current interpretations of the parametric Budyko equations and points out that there is a fundamental contradiction between the parametric equations and the postulate that solutions to the "original" Budyko hypothesis, $\bar{E} = f(\bar{P}, \overline{E_0})$, are homogeneous. This leads Sposito (2017) to state, "It must be concluded, therefore, that the current physical interpretation of the MCY and Fu model parameters [n and w, respectively] may be spurious." (page 12 of Sposito (2017), a statement we support.

Regarding how this work provides an advance over previous work, prior to this technical note, the parametric Budyko equations have only been inverted numerically; providing analytical expressions for this inversion is a novel advancement in the understanding of $n$ and $w$ and provides the general relationship between the catchment-specific parameters and biophysical features. An important interpretation of this relationship is that values for $n$ and $w$ can only be computed if $\bar{P}$, $\overline{E_0}$, and $\bar{E}$ are known *a priori* (obtained either from empirical data or models). Thus, any dependence $n$ and $w$ have on biophysical features must be through the dependence of $\bar{P}$, $\overline{E_0}$, and $\bar{E}$ on those same features. We address this concept briefly in the original manuscript but propose further edits to the abstract, introduction, and discussion and conclusions sections to make this point more strongly (see response to **Reviewer 1, Comment 3**).

Finally, we note that the many previously proposed statistical relationships between catchment biophysical features and $n$ or $w$ previously proposed, e.g., (Yang et al., 2007;Donohue et al., 2012;Yang et al., 2009;Shao et al., 2012;Li et al., 2013;Xu et al., 2013;Cong et al., 2015;Yang et al., 2016;Zhang et al., 2018;Abatzoglou and Ficklin, 2017;Xing et al., 2018;Zhao et al., 2020;Ning et al., 2020b;Ning et al., 2020a;Li et al., 2020b;Li et al., 2020a;Zhang et al., 2019;Ning et al., 2019;Bai et al., 2019;Ning et al., 2017) are all incomplete, special-case approximations of the explicit expressions derived in our manuscript. Such special-case approximations are limited by the specific data and regression models used in their development. The novel contribution of the analytical expressions we derived is that $\overline{E_0}$, $\bar{P}$, and $\bar{E}$ can be first expressed in terms of biophysical features (either theoretically or empirically) and then substituted into Equation 14 to understand the explicit relationships among $n$, $w$, and proposed biophysical features.

**Comment 4:**
**My other point is that, while the authors provide an analytical solution to how to define w and n, I am unsure of what problem it is that they are solving. They indicate the problem has been stated in the literature (L62): "Notably, there has not been an analytical derivation illustrating how $n$ and $w$ relate to biophysical features, though the importance of doing so has been noted many times." and (L169): ": : :the literature-identified need of an analytical expression." However, the motivation behind trying to**

predict a catchment's parameter value is to be able to use Budyko to make predictions about E and Q based on Ep and P – that is, in ungauged catchments. The analytical definitions of w and n are given here as functions of E and so cannot address the need most of the literature is trying to address, which is prediction in ungauged catchments.

Response 4:
We agree with the reviewer's assessment that the utility of explicit expressions for $n$ and $w$ (Equation 14, page 4, line 85) is not for prediction of $\bar{E}$ and $\bar{Q}$ in ungauged catchments. Rather, these explicit expressions illustrate how the catchment-specific parameter can depend on biophysical features, but only through the dependence of $\overline{E_0}$, $\bar{P}$, and $\bar{E}$ on those same features. One of the implications of this dependence is that if only $\overline{E_0}$ and $\bar{P}$ are known for an ungauged catchment, $n$ or $w$ cannot be determined (Reaver et al. (2020). Additionally, the complexity and highly non-linear nature of relationship between $n$ or $w$ and $\overline{E_0}$, $\bar{P}$, and $\bar{E}$ highlights how the many studies purporting to develop explicit expressions for $n$ or $w$ in terms of biophysical features arrive at such widely divergent results, both in the functional forms of the relationships and what biophysical features are included.

**Comment 5:**
**Further, the need stated in the literature calls for a biophysical understanding of the parameters. The authors claim their solution ": : :.thus fulfills the literature-identified need of an analytical expression for $n$ and $w$ in terms of biophysical features." . While they have provided an analytical solution I am not convinced it provides a solution that is any more connected to biophysical features any more than the original formulations are. Another way of saying this is, the solution doesn't provide any greater biophysical understanding of the meaning of w or n than previously existed, nor does it make Budyko any more useful. Again, how does this then address the need to be able to predict w and n ungauged catchments? I think that the authors need to rethink what is the question they are trying to address and ensure it represents an advance in the use of Budyko to make hydrological predictions.**

Response 5:
Through our derivation, we provide a greater understanding of how $n$ and $w$ are connected to catchment biophysical features (i.e., through the dependence of $\overline{E_0}$, $\bar{P}$, and $\bar{E}$ on those same features). It is true that this result does not improve the utility of the parametric Budyko equations, but it does explain how the catchment-specific parameter is connected to biophysical features, and thus addresses the calls for a biophysical understanding of $n$ and $w$. While this may not be the resolution that the literature calls for (i.e., an analytical expression for $n$ and $w$ in terms of biophysical features), it is nevertheless the resolution obtained through a logical and careful consideration of the parametric Budyko equations. To make this point more explicitly, we propose to add the following text to the final paragraph of the discussion and conclusions section (after the proposed text in our response to **Reviewer 1, Comment 3**):

*"While this result does not improve the utility of the parametric Budyko equations, it explains how the catchment-specific parameter is connected to biophysical features, and thus addresses the calls for a better biophysical understanding of n and w. While this may not be the intended resolution called for in the literature (i.e., an analytical expression for n and w in terms of biophysical features), it is nevertheless the outcome obtained through a logical and careful consideration of the parametric Budyko equations."*

To close, we note that our conclusions about the nature of $n$ and $w$ emerged from a genuine interest in the parametric Budyko equations and a careful study of the catchment hydrology literature in an attempt

to improve the biophysical understanding of the catchment-specific parameters. We wholeheartedly agree that these explicit expressions do not make the parametric Budyko equations more useful for making hydrological predictions, however, we emphasize that their underlying "uselessness" is due to the fact that the parametric Budyko equations themselves are not useful for making hydrological predictions, a theme we elaborate on in detail in the companion research article (Reaver et al. 2020).

**References:**

[revised manuscript text omitted]